# The Decadal Climate Prediction Project (DCPP) contribution to CMIP6

George J. Boer[1], Douglas. M. Smith[2], Christophe Cassou[3], Francisco Doblas-Reyes[4], Gokhan Danabasoglu[5], Ben Kirtman[6], Yochanan Kushnir[7], Masahide Kimoto[8], Gerald A. Meehl[5], Rym Msadek[3,12], Wolfgang A. Mueller[9], Karl Taylor[10], Francis Zwiers[11], Michel Rixen[13], Yohan Ruprich-Robert[14], R. Eade[2]

[1]Canadian Centre for Climate Modelling and Analysis, Environment and Climate Change Canada, Victoria BC, Canada
[2]Met Office, Hadley Centre, Exeter, UK
[3] Centre National de la Recherche Scientifique (CNRS)/CERFACS, CECI,,UMR 5318 Toulouse, France
[4]Institució Catalana de Recerca i Estudis Avançats (ICREA) and Barcelona Supercomputing Center (BSC-CNS), Spain
[5]National Center for Atmospheric Research (NCAR), Boulder CO, USA
[6]Rosenstiel School of Marine and Atmospheric Science, University of Miam FLi, USA
[7] Lamont Doherty Earth Observatory, Palisades NY, USA
[8] Atmosphere and Ocean Research Institute, University of Tokyo, Japan
[9] Max-Planck-Institute for Meteorology, Hamburg, Germany
[10]Program for Climate Model Diagnosis and Intercomparison (PCMDI), Lawrence Livermore National Laboratory, Livermore, CA, USA
[11]Pacific Climate Impacts Consortium, Victoria BC, Canada
[12] Geophysical Fluid Dynamics Laboratory/NOAA, Princeton, NJ, USA
[13]World Climate Research Programme. Geneva, Switzerland.
[14]Atmosphere and Ocean Sciences, Princeton University, Princeton, NJ, USA

*Correspondence to*: G.J. Boer (george.boer@canada.ca), D. M. Smith (doug.smith@metoffice.gov.uk)

**Abstract.**

The Decadal Climate Prediction Project (DCPP) is a coordinated multi-model investigation into decadal climate prediction, predictability, and variability. The DCPP makes use of past experience in simulating and predicting decadal variability and forced climate change gained from CMIP5 and elsewhere. It builds on recent improvements in models, in the reanalysis of climate data, in methods of initialization and ensemble generation, and in data treatment and analysis to propose an extended comprehensive decadal prediction investigation as a contribution to CMIP6 (Eyring et al., 2016) and to the WCRP Grand Challenge on Near Term Climate Prediction (Kushnir et al., 2016). The DCPP consists of three Components. Component A comprises the production and analysis of an extensive archive of retrospective forecasts to be used to assess and understand historical decadal prediction skill, as a basis for improvements in all aspects of end-to-end decadal prediction, and as a basis for forecasting on annual to decadal timescales. Component B undertakes ongoing production, analysis and dissemination of experimental quasi-real-time multi-model forecasts as a basis for potential operational forecast production. Component C involves the organization and coordination of case studies of particular climate shifts and variations, both natural and naturally forced (e.g. the "hiatus", volcanoes), including the study of the

mechanisms that determine these behaviours. Groups are invited to participate in as many or as few of the Components of the DCPP, each of which are separately prioritized, as are of interest to them.

The Decadal Climate Prediction Project addresses a range of scientific issues involving the ability of the climate system
to be predicted on annual to decadal timescales, the skill that is currently and potentially available, the mechanisms involved in long timescale variability, and the production of forecasts of benefit to both science and society.

## 1 Introduction

The term "decadal prediction", as used here, encompasses predictions on annual, multi-annual to decadal timescales. The
possibility of making skillful forecasts on these timescales, and the ability to do so, is investigated by means of predictability studies and retrospective predictions (hindcasts) made using the latest generation of climate models. Skillful decadal prediction of relevant climate parameters is a Key Deliverable of the World Climate Research Programme (WCRP) Grand Challenge of Near Term Climate Prediction.

The Decadal Climate Prediction Panel, in conjunction with the Working Group on Seasonal to Interannual Prediction (WGSIP) and the Working Group on Coupled Modelling (WGCM ) is coordinating the scientific and practical aspects of the Decadal Climate Prediction Project (DCPP) which will contribute to the 6[th] Coupled Model Intercomparison Project (Eyring et al., 2016). The CMIP6 website (http://www.wcrp-climate.org/wgcm-cmip/wgcm-cmip6) contains information on CMIP6, including links to forcing information and data treatment. The DCPP website (http://www.wcrp-climate.org/dcp-overview)
contains up-to-date information on the DCPP and related issues.

Predictability is a feature of a physical and/or mathematical system which characterizes "its ability to be predicted" as indicated, for instance, by the rate at which the trajectories of initially close states separate. Predictability may be estimated from models although with the proviso that such indications depend on the model on which they are based and do not
necessarily fully represent the behaviour of the physical climate system. Predictability studies, used with care, can give an indication as to where, under what circumstances, and the level of confidence with which it may be possible to predict various climate parameters on timescales from seasons to decades.

Forecast skill, on the other hand, is measured by comparing initialized forecasts with observations and indicates the "ability
to predict" the actual evolution of the climate system. A forecast is essentially useless unless there is some indication of its expected skill. A sequence of retrospective forecasts (known as "hindcasts") made with a single model, or preferably multiple models, can provide historical skill measures as well as estimates of predictability. The forecasts also provide information, together with targeted simulations, for understanding the physical mechanisms that govern climate variation and this is important for the science as well as for engendering confidence in the forecasts.

The evolution of the forecast and observed variables of the physical climate system are a combination of externally forced and internally generated components, both of which are important on annual to decadal timescales. The externally forced components are the result of changes in greenhouse gases, anthropogenic and volcanic aerosols, variations in solar irradiance and the like. Examples of internally generated variability include the El Niño Southern Oscillation (ENSO), important on annual timescales, and the multi-year to multi-decadal variations in both the Pacific and Atlantic Oceans. Decadal predictions encompass aspects of both an initial value and a forced boundary value problem as indicated in Figure 1. It is important for successful decadal prediction that both the externally forced and internally generated components of the system are initialized and it is also useful to diagnose their individual contribution to the skill of the hindcasts and forecasts.

The DCPP's extensive archive of annual, multi-annual to decadal climate hindcasts and results of targeted experiments will support improved understanding of the mechanisms underlying forced and naturally occurring climate variability. The information generated by the DCPP can provide a basis for socially relevant operational climate predictions on annual to decadal timescales. These results will be of interest generally as well as to international organizations such as the Global Framework for Climate Services (GFCS) and the WMO Commission for Basic Systems (CBS).

## 2 Decadal prediction and CMIP5

While long-term climate simulations have been investigated for some time, the fifth Coupled Model Intercomparison Project (CMIP5; Taylor et al., 2012) represents one of the first attempts at a coordinated multi-model initialized decadal forecasting experiment as illustrated in Figure 2. Results based on the hindcasts from the CMIP5 experiments have been reported in the literature and have contributed to Chapter 11(Kirtman et al, 2013) of the Intergovernmental Panel on Climate Change (IPCC) fifth assessment report (IPCC 2013) entitled Near-term Climate Change: Projections and Predictability. These comparatively early results indicate that there is skill in predicting the annual mean temperature evolution for a number of years into the future (e.g., Doblas-Reyes et al. 2013). The upper panels of Figure 3 plot the correlation skill of the year 1 and year 2 forecasts and the year 2-5 average forecast for surface air temperature. The impact of initialization, based on differences between uninitialized historical simulations and initialized decadal predictions, is plotted in the lower panels. The results are based on the output of 5 forecast models participating in CMIP5 (CanCM4, GFDL, MPI-ESL-LR, MIROC5, HadCM3) and the HadCM3 PPE from the ENSEMBLES project. Similar results for precipitation are available but show considerably less skill at this stage. The expectation is that the improvements in the forecast systems participating in CMIP6 will lead to improved skill for this important parameter (Smith et al., 2012). The results in Figure 3 are based on earlier models and approaches but it is clear that predictions of surface air temperature have considerable skill for a number of years and for multi-year averages. The enhancement of skill due to the initialization of the forecasts is greatest in the first few years and

for particular regions such as the North Atlantic (as discussed under Component C) and becomes less so at longer forecast ranges where skill is provided mainly by the externally forced component.

This behaviour is seen also in Figure 4 where the global average of the correlation skill for surface air temperature from a single model is plotted. The orange curve indicates the overall correlation skill associated with the prediction of both forced and internally generated components of variability. While the separation is approximate, the blue curve estimates the skill associated with the initialization of the internally generated component and the difference between the curves indicates the skill associated with the forced component. The skill of the initialized internally generated component displays classical forecast behaviour and declines toward zero as the forecast progresses. The externally forced component, on the other hand, maintains skill at longer forecast times. The result is that the overall skill of decadal forecasts does not decline to zero but plateaus or even increases as forecast range increases. Finally, Figure 4 also plots an estimate of "potential skill" which is the skill that the model obtains when predicting its own evolution. To the extent that the model suitably reflects the behaviour of the actual system this at least suggests that there may be additional skill that could be accessed by the improved forecasting systems that will be used in the DCPP.

## 3 The DCPP and CMIP6

The approach taken in the DCPP contribution to CMIP6 differs in some respects from that in CMIP5 although both climate simulations and decadal hindcasts are again important components. The DCPP contribution is a CMIP6-endorsed model intercomparison project which consists of three Components, each of which comprises a central "core" and additional desirable, but less central, experiments and integrations. Terminology has changed slightly compared to CMIP5 in Figure 1 with core experiments now denoted as "Tier 1" and so on for the other tiers. The experience gained in CMIP5 and the subsequent improvements made in forecast systems make it timely to revisit an improved and extended decadal prediction component for CMIP6.

The lessons learned from the CMIP5 decadal prediction experiments have been incorporated into the design of the DCPP. Differences in the CMIP6 experimental protocol compared to that of CMIP5 include more frequent hindcast start dates and larger ensembles of hindcasts for each start date intended to provide robust estimates of skill (e.g. Sienz et al. 2016), the addition of ongoing quasi-operational experimental decadal forecasts (Smith et al. 2013), and the addition of targeted experiments to provide insight into the physical processes affecting decadal variability and forecast skill (e.g. Ruprich-Robert et al. 2016).

The three Components of the DCPP are:

- *Component A, Hindcasts:* the design and organization of a coordinated decadal prediction (hindcast) component of CMIP6 in conjunction with the seasonal prediction and climate modelling communities and the production of a comprehensive archive of results for research and applications
- *Component B, Forecasts:* the ongoing production of experimental quasi-operational decadal climate predictions in support of multi-model annual to decadal forecasting and the application of the forecasts to societal needs
- *Component C, Predictability, mechanisms, and case studies:* the organization and coordination of decadal climate predictability studies and of case studies of particular climate shifts and variations including the study of the mechanisms that determine these behaviours

Components A and B are directed toward the production, analysis and application of annual, multi-annual to decadal forecasts. A major output of these Components is a multi-model archive of retrospective and real-time forecasts, which will serve as a resource for the analysis, understanding, and improvement of near-term climate forecasts and forecasting techniques and for their potential application (e.g. Asrar and Hurrell, 2013, Caron, et al., 2015).

Component C proposes targeted investigations which seek to understand some of the mechanisms that produce long timescale variability in the climate system and that support successful predictions of both internally generated and externally forced climate variability. Mechanisms investigated via targeted simulations include aspects and effects of Atlantic Multidecadal Variation (AMV, also referred to as the Atlantic Multidecadal Oscillation or AMO) and the Interdecadal Pacific Variation (IPV, similarly the IPO) as well as volcanic effects on prediction and predictability. Many scientific and practical questions are involved. The understanding of the physical processes that govern the long timescale predictability of the climate system is vital for improving decadal predictions and gaining further confidence in forecasts.

The DCPP contribution to CMIP6 represents an evolution of the design of the CMIP5 decadal prediction effort but also, and perhaps more importantly, embodies the evolution and improvement of the components of end-to-end hindcast/forecasting systems. The research and development efforts contributing to the DCPP include improvements in the analysis of the observations available for initializing forecasts (e.g. Chapters 2-4, IPCC 2013), in methods of initializing models and of generating ensembles of initial conditions(e.g. Balmaseda et al., 2015 and others in this Special Issue) in the representation of atmospheric, oceanic and terrestrial components of the models used in the production of the forecasts and in their coupling (e.g. Chapter 9, IPCC 2013), in methods of post processing the forecasts including new approaches to bias adjustment, to calibration and multi-model combination of the forecasts, and in production and application of probabilistic decadal forecasts (e.g. Troccoli et al., 2008, Jolliffe and Stephenson, 2011). Since many of these topics have been treated fairly recently in the IPCC Fourth Assessment Report (IPCC 2013) for instance, we do not attempt to review them here. One of the goals of the DCPP is to encourage new methods and approaches to decadal forecast production rather than to specify rigid procedures.

## 4 A multi-system approach

The DCPP represents a "multi-system" approach to climate variation and prediction in which the basic experiments are specified but the details of the implementation are not. The reason for this resides in the uncertainty inherent in any climate prediction or simulation. The DCPP does not specify the data or the methods to be used to initialize forecasts (e.g. full field or anomaly initialization) or how to generate ensembles of initial conditions. The assumption is that the differences in initializing data sets sample the uncertainty that the ensemble of initial conditions is meant to represent. The consequences of these uncertainties in initial conditions are expressed both within and across model results. Differences in model resolution and physical formulation also give rise to differences in results which are a reflection of uncertainty. A "multi-model forecast" (better a multi-system forecast) combines the results from forecast systems which partake of diverse models and methods, each of which represent the "best efforts" of the modelling groups involved. The overall result is increased skill (e.g. IPCC Chapter 11, DelSole et al., 2014). Differences between models may also be analyzed to understand how different formulations affect the skill of the forecasts.

As has been the case for weather and seasonal forecasting (e.g. Bauer et al., 2015, MacLachlan et al., 2015) continued improvement in each of the components of a decadal forecasting system is expected to yield improvement in decadal prediction skill. These considerations apply also to improvements in the simulation and understanding of climate system behaviour as represented by the sequence of CMIP efforts culminating in CMIP6 (Eyring et al., 2016).

## 5 Analysis of results

The scientific analysis of the DCPP is predicated on its broad multi-system approach. In particular, the extensive archive of multi-system results are a resource for the analysis community and many novel and innovative analyses will be undertaken based on the availability of these data. The improvements in the design of the Component A hindcast experiment, the broader participation compared to CMIP5, and the augmented archive of results provides the basis for many types of analyses. The most obvious analysis results for Component A hindcasts are measures of historical forecasts skill on annual, multi-annual to decadal forecast ranges for each system and, ultimately, for an optimum combination of these results into a multi-system forecast. The skill of the initialized forecasts compared to the results of historical climate simulations is a measure of the impact of initialization and is certainly of interest. These analyses require the bias correction of the forecasts, a version of which is as discussed in Appendix E. No one measure can convey all of the verification information available from a set of hindcasts/forecasts (e.g. Jolliffe and Stephenson, 2011) nevertheless there are basic measures that can be used as suggested in Goddard et al., (2013) and in the Standard Verification System for Long-range Forecasts from the World Meteorological Organization (Graham et al., 2011).

An archive of ensembles of hindcasts also permits estimates of the predictability, as opposed to the forecast skill, of the system and of its components (e.g. Boer et al.2013, Hawkins et al, 2016). To the extent that the model (or multi-model combination) successfully reproduces climate system behaviour, predictability results indicate where geographically, and for which variables, there may be the possibility of improving the forecast system. The overall behaviour of the forecasts and the associated predictability statistics can also reveal aspects of the mechanisms involved in governing predictability and skill as well as deficiencies in aspects of model behaviour that mitigate against skill (e.g. Eade et al., 2014).

Component B will ultimately make use of the results of Component A and together will provide research support for the eventual production of operational decadal predictions. The DCPP is an essential part of the recently established WCRP Grand Challenge of Near Term Climate Prediction (Kushnir et al., in preparation). The Grand Challenge goals include the adoption of standards, verification methods and guidance for decadal predictions, the WMO recognition for operational decadal prediction and eventually the issuance of a real-time Global Decadal Climate Outlook each year.

There are many studies of important long timescale behaviours affecting the climate system including the so-called "hiatus" in global warming and the coupled AMV and IPV processes (see the entries in many Chapters of IPCC 2013). Despite these studies the processes governing these mechanisms, and their teleconnected effects, are not fully understood. The analysis of Component C archives will bring multi-system results to bear on the understanding of these mechanisms and on their effects on predictability and forecast skill. Analysis methods are being developed (e.g. Section 11 and references therein) where the existence of a broad archive of results offers the opportunity for new and innovative approaches.

Although episodic and of differing magnitudes, volcanic eruptions have effects on climate and on decadal predictability and skill which are of interest and importance. These are investigated in the multi-system context in Component C in collaboration with VolMIP and using the analyses methods they suggest as well as the general approaches to skill and predictability applied to the other DCPP Components.

**6. DECK and CMIP6 historical simulations.** The DCPP is unique in bringing together researchers from communities with expertise in seasonal to interannual prediction as well as climate simulation. For climate models, control and sensitivity experiments are a backdrop to climate change simulations and most models used in the DCPP will also participate in other aspects of CMIP6 and will have performed DECK and 20th century climate change integrations as suggested by CMIP6. Climate simulations, both equilibrium and historical, compare ensemble and time averaged results to the model's equilibrium pre-industrial climate with results that are partially characterized by the models sensitivity to increasing $CO_2$. A decadal hindcast or forecast, by contrast, is characterized by its ability to reproduce the details of system evolution on timescales up to a decade. Results depend on the initial observation-based state, which includes system forced climate change to that point, as well as the state of the unforced internally generated climate variability. An important aspect of the

analysis of results is the comparison of the forecasts with the results of historical climate change simulations with the difference representing the added information available from the initial conditions. This is another motivation for including an ensemble of historical simulations as one tier of the Component A specifications.

Comparing decadal forecast to observations provides a different, and in some ways richer, characterization of model behaviour than is possible with the DECK results alone. For instance, Ma et al., (2014) analyse the time scale over which systematic errors develop thus yielding insights into their origin. As well, as forecasts evolve, they lose initial condition information and approach a forced climate state giving information also on this behaviour. For these reasons, while the DCPP strongly encourages participants to perform the DECK simulations it is recognized that this may not be feasible for all
groups (those proposing to use high resolution models for prediction for instance). It is not intended that the DECK requirements should bar DCPP participation in these special cases.

### 7. Interactions with other MIPs

Interactions with other CMIP6 MIPs include: a common approach to some IPV and AMV experiments in GMMIP which
will contribute to both; coordinated experiments with VolMIP with and without major volcanic forcing; outputs of DCPP hindcasts for DynVar; and the ensemble of DCPP hindcasts and simulations as contributions to DAMIP, SolarMIP and ScenarioMIP,

### 8. Participation

Groups are invited to participate in as many or as few of the Components, each of which are separately "tiered", as are of interest to them. The number of years of integration associated with the different Tiers of each of the Components and sub-Components is listed in Table 1 where the Tier 1 experiments are shaded in yellow. Groups are invited to consider also the Tier 4 experiments but these are expected to be of interest to fewer groups. It is hoped that most groups will participate in the Tier 1 experiments associated with at least one of the Components but it is not expected that all groups will participate in all
experiments or tiers.

### 9. DCPP Component A: A multi-year multi-model decadal hindcast experiment

The decadal hindcast component of CMIP follows the example of other coordinated experiments as a protocol-driven multi-model multi-national project with data production and data sharing as integral components.

**The Goals** of the decadal hindcast component of CMIP include:
- the promotion of the science and practice of decadal prediction (forecasts on timescales up to and including 10 years)

- the provision of information potentially useful for the IPCC WG1 AR6 assessment report and other studies and reports on climate prediction and evolution
- the production and retention of a multi-year multi-model collection of decadal hindcast data in support of climate science and of use to the Global Framework for Climate Services (GFCS) and other organizations

**Scientific aspects** of the DCPP to which Component A can contribute include:

- a system view (data; analyses; initial conditions; ensemble generation; models and forecast production; post processing and assessment) of decadal prediction
- the investigation of broad questions (e.g. sources and limits of predictability, current abilities with respect to decadal prediction, potential applications, …)
- the provision of benchmarks against which to compare improvements in forecast system components and their contribution to prediction quality
- information on processes and mechanisms of interest (e.g., the hiatus, climate shifts, Atlantic meridional overturning circulation (AMOC), etc.) in a collection of hindcasts

**Practical aspects** of Component A include:

- the coordination of efforts based on agreed experimental structures and timelines in order to promote research, intercomparison, multimodel approaches, applications, and new research directions
- a contribution to the development of infrastructure, in particular a multi-purpose data archive of decadal hindcasts useful for a broad range of scientific and application questions and of benefit to national and international climate prediction and climate services organizations.

**The basic elements** of Component A are:

- a coordinated set of multi-model multi-member ensembles of retrospective forecasts initialized each year from 1960 to the present
- the resulting archive of forecast results generally and readily available to the scientific and applications communities via the Earth System Grid Federation (ESGF)

**Consultation and timing** for Component A:

- The proposed timing for Component A generally follows that outlined for CMIP6 (Eyring et al. 2016). In particular, the availability of historical forcing and future scenario information are key to DCPP timing.

Details of the proposed Component A decadal prediction hindcasts are listed in **Appendix A.**

## 10. DCPP Component B: Experimental real-time multi-model decadal predictions

The real-time decadal prediction component of the DCPP also follows the example of other coordinated experiments as a protocol-driven multi-model multi-national project with data production and data sharing as integral components. The WMO structure already in place for seasonal forecasts is an example. Forecasts and verification statistics will be made available on

the ESGF as part of CMIP6.  Current efforts in quasi-real time annual and multi-annual predictions are being undertaken by individual groups, are collected at the UK Met Office (http://www.metoffice.gov.uk/research/climate/seasonal-to-decadal/long-range/decadal-multimodel), and provide the basis of a multi-model prediction effort (Smith et al. 2013). An example of such a multi-model quasi-real-time prediction is shown in Figure 5. Results from the newer forecasting systems employed in Component A will be incorporated as they become available and are expected to improve these quasi-operational forecasts. At some later time the WMO may designate "Lead Centres" to collect forecast and verification data in order to produce an operational multi-model real-time forecast together with an assessment of performance. A demonstrated ability to produce skillful real-time multi-annual forecasts will be a contribution to the GFCS and will fill a gap between seasonal predictions and long term climate change projections.

**Goals:**

- as for Component A but with the added dimension that the goals apply to quasi-operational real-time multi-model decadal predictions

**Scientific aspects**

- the assessment of decadal predictions of key variables including surface temperature, precipitation, mean sea level pressure, AMV, IPV, Arctic sea ice, the North Atlantic Oscillation (NAO), and tropical storms
- the assessment of uncertainties and the generation of a consensus forecast
- the assessment of decadal predictions and associated climate impacts of societal relevance

**Basic elements**

- an ongoing coordinated set of multi-model multi-member ensembles of real-time forecasts, updated each year
- an associated hierarchy of data sets of results generally and readily available to the scientific and applications communities including National Meteorological and Hydrological Services and Regional Climate Centres.

Details of the proposed Component B real-time decadal decadal prediction component are listed in the **Appendix B.**

## 11 DCPP Component C: Predictability, Mechanisms and Case Studies

The climate system varies on multiple timescales which may be studied using physically based and statistical models. Diagnostic studies investigate climate system behaviour inferred indirectly from a long series of observations and/or model simulations. Prognostic studies investigate the behaviour of models when initial conditions or model features such as physical parameterizations, numerical treatments or forcings are perturbed. The mechanisms involved in the long timescale behaviour of the climate system are of great interest as they underpin the inherent predictability of the system that governs forecast skill.

Case studies are hindcasts which focus on a particular climatic event and the mechanisms and impacts involved. These are typically hindcast studies of an observed event although they can include particular kinds of events seen in model integrations (variations of AMOC and the associated variation of the North Atlantic sea surface temperatures (SSTs) in models are an example). Studies of the skill with which a particular event (e.g. the hiatus, climate shift, an extreme year, etc.)
can be forecast and the mechanisms which support (or perhaps make difficult) a skilful prediction are all of interest.

The DCPP and the CLIVAR Decadal Climate Variability and Predictability (DCVP) focus group are proposing coordinated multi-model investigations of a limited number of mechanism/predictability/case studies believed to be of broad interest to the community. Two research areas are the current foci of Component C. They are:
• Hiatus+: this is used as shorthand to indicate investigations into the origins, mechanisms and predictability of long timescale variations in both global mean surface temperature (and other variables) and regional imprints including periods of both enhanced global warming and cooling with a focus on the most recent slowdown that began in the late 1990s.
   • Volcanoes in a prediction context: an investigation of the influence and consequences of volcanic eruptions on
decadal prediction and predictability
Full details of the proposed experiments are given in **Appendix C.**

The proposed experiments in Table C1 of Appendix C are intended to discover how models respond to imposed slowly evolving SST anomalies in the Atlantic and the Pacific, which are perceived as originating in ocean heat content or heat
transport convergence anomalies. The questions at issue are the consistency of models' responses to these SSTs and the pathways through which the responses are expressed throughout the ocean and atmosphere. The experiments are expected to illuminate model behaviour on decadal time scales and possible mechanistic links to retarded and accelerated global surface temperature variations and regional climate anomalies. In other words, to what extent can modulations of global mean surface temperatures be attributed to ocean heat content variations, what are the respective roles of Atlantic and Pacific SST
anomalies in these changes, and to what extent can we attribute decadal climate anomalies at regional scale (particularly over land) to the patterns of Atlantic Multidecadal Variability (AMV) and Pacific Decadal Variability (PDV) sea surface temperature that are illustrated in Figures 6 and 7. These experiments also address the interrelationships between the AMV and PDV shifts and the mechanisms at play.

A second set of Component C experiments in Table C2 of Appendix C investigates the predictability of the mid-1990s warming of the Atlantic subpolar gyre, and its impacts on climate variability. Some CMIP5 decadal hindcasts successfully predicted this event (Robson et al. 2012, Yeager et al. 2012, Msadek et al. 2014) together with some aspects of associated climate impacts (Robson et al. 2013, Smith et al. 2010). The proposed experiments will investigate in more detail the role of initialization of the Atlantic subpolar gyre. Analysis of these experiments will include assessment of the role of the Atlantic

Meridional Overturning Circulation (AMOC) in the subpolar gyre warming, and the impact of the subpolar gyre on the AMV pattern and associated climate impacts, including rainfall over the Sahel, Amazon, US and Europe, and Atlantic tropical storms.

The final set of Component C experiments in Table C3 of Appendix C are jointly proposed with VolMIP (Zanchettin et al. 2016) and are directed toward an understanding of the effects of volcanoes on past and potentially on future decadal predictions. Removing the forcing due to major volcanic eruptions from hindcasts during which they occurred and introducing volcanic forcing into forecasts during which no volcano occurred will allow estimates of the impact on skill to be made (e.g. Maher et al., 2015, Meehl et al., 2015, Timmreck et al., 2016). Comparing the effects of the same eruption in
hindcasts and forecasts also allows the impact of the background climate state to be assessed. In addition to assessing the radiative effects arising from the aerosol loading in the stratosphere, an important aspect of the analysis of these experiments will be to investigate subsequent dynamical responses including, for example, those involving the NAO and ENSO.

    Participants are invited to undertake as many or as few of the Component C experiments as are of interest to them. Please see
the Notes at the end of Appendix C for additional details on the Component C experimental protocol.

**12 Concluding comments**

    The DCPP is unique in bringing together researchers from communities with expertise in seasonal to interannual prediction
(as represented by WGSIP), climate simulation (as represented by WGCM), and decadal variability and predictability in general (as represented by CLIVAR). The models used and approaches taken represent to varying degrees the interests and abilities of these communities.

    For climate models, control and sensitivity experiments are a necessary backdrop to climate change simulations. Most
models used in the DCPP will also participate in other aspects of CMIP6 and will have performed climate integrations as well as other simulations and MIP experiments. The data retained for these studies provides information on forced responses and the statistics of internal variability which are important for DCPP-related studies of many different aspects of decadal variability and prediction. The forecasting aspect of DCPP encourages emphasis on methods of initializing models, generating ensembles of forecasts and, especially, on assessing results against observations. The two approaches represent
complementary views for the understanding and prediction of forced and internally generated climate variations. The tiered set of retained data for the DCPP is intended to assist in the evaluation and analysis of DCPP results, but groups are encouraged to retain additional data relevant to other MIPs if possible.

We believe that the Decadal Climate Prediction Project represents an important evolutionary advance from the CMIP5 decadal prediction component and addresses an integrated range of scientific issues broadly characterized as the ability of the system to be predicted on decadal timescales, the currently available skill, the mechanisms that control long timescale variability, and the ongoing production of forecasts of potential benefit for both science and societal applications. This will be a major resource to support the WCRP's new Grand Challenge of Near Term Climate Prediction and an important asset for the development of climate services on time scales relevant to a wide range of users .

## 13 Data Availability

The model output from DCPP hindcasts, forecasts, and targeted experiments described in this paper will be distributed through the Earth System Grid Federation (ESGF) with digital object identifiers (DOIs) assigned. The list of requested variables, including frequencies and priorities, is given in Appendix D and has been submitted as part of the "CMIP6 Data Request Compilation". As in CMIP5, the model output will be freely accessible through data portals after a simple registration process that is unique to all CMIP6 components. In order to document CMIP6's scientific impact and enable ongoing support of CMIP, users are requested to acknowledge CMIP6, the participating modelling groups, and the ESGF centres (see details on the CMIP website). Further information about the infrastructure supporting CMIP6, the metadata describing the model output, and the terms governing its use are provided by the WGCM Infrastructure Panel (WIP). Links to this information may be found on the CMIP6 website and is discussed in the WIP contribution to this Special Issue. Along with the data itself, the provenance of the data will be recorded, and DOI's will be assigned to collections of output so that they can be appropriately cited. This information will be made readily available so that research results can be compared and the modelling groups providing the data can be credited.

The WIP is coordinating and encouraging the development of the infrastructure needed to archive and deliver the large amount of information generated by CMIP6. Datasets of natural and anthropogenic forcing information are required for the DCPP hindcasts, forecasts, and simulations as defined for the CMIP6 historical simulations and ScenarioMIP. These datasets are described in separate contributions to this Special Issue and will be made available through the ESGF with version control and DOIs assigned.

**Table1. DCPP Experiments**

| | Expmt | experiment_id | Tier | Years | Description |
|---|---|---|---|---|---|
| Component A: Decadal Hindcasts | A1 | dcppA-hindcast | 1 | 3000 | Five year hindcasts every year from 1960. Note that the first forecast year is 1961 from initialization toward the end of 1960. |
| | A2.1 | | 2 | 3000 | Extend A1 hindcast duration to 10 years |
| | A2.2 | dcppA-historical | 2 | 1700 | Ensemble of uninitialized historical/future simulations |
| | A2.3 | dcppA-assim | 2 | (60-600) | Ensemble of "assimilation" run(s) (if available). These are simulations used to incorporate observation-based data into the model in order to generate initial conditions for hindcasts. They parallel the historical simulations and use the same forcing. The number of years depends on the number of independent assimilation runs. |
| | A3.1 | dcppA-hindcast | 3 | 300m | Increase ensemble size by $m$ for A1 |
| | A3.2 | | 3 | 300m | Increase ensemble size by $m$ for A2.1 |
| | A4.1 | dcppA-hindcast-niff | 4 | 3000 | As A1 but no forcing information from the future (niff) with respect to the hindcast. Forcing from persistence or other estimate. |
| | A4.2 | dcppA-historical-niff | 4 | 3000 | As A4.1 but initialized from historical simulations |
| Component B: Decadal Forecasts | B1 | dcppB-forecast | 1 | 50 | Ongoing near real-time forecasts |
| | B2.1 | | 2 | 5m | Increase ensemble size by $m$ for B1 |
| | B2.2 | | 2 | 50 | Extend forecast duration to 10 years for B1 |
| Component C: Hiatus+ | C1.1 | dcppC-atl-control | 1 | 250 | Idealized Atlantic control |
| | C1.2 | dcppC-amv-pos | 1 | 250 | Idealized impact of AMV+ |
| | C1.3 | dcppC-amv-neg | 1 | 250 | Idealized impact of AMV- |
| | C1.4 | dcppC-pac-control | 1 | 100 | Idealized Pacific control |
| | C1.5 | dcppC-ipv-pos | 1 | 100 | Idealized impact of IPV+ |
| | C1.6 | dcppC-ipv-neg | 1 | 100 | Idealized impact of IPV- |
| | C1.7 | dcppC-amv-ExTrop-pos dcppC-amv-ExTtrop-neg | 2 | 500 | Idealized impact of extratropical AMV+ and AMV- |
| | C1.8 | dcppC-amv-Trop-pos dcppC-amv-Trop-neg | 2 | 500 | Idealized impact of tropical AMV+ and AMV- |
| | C1.9 | dcppC-ipv-NexTrop-pos dcppC-ipv-NexTtrop-neg | 2 | 200 | Idealized impact of northern extratropical IPV+ and IPV- |
| | C1.10 | dcppC-pac-pacemaker | 3 | 650 | Pacemaker Pacific experiment |

| | | | | | |
|---|---|---|---|---|---|
| | C1.11 | dcppC-atl-pacemaker | 3 | 650 | Pacemaker Atlantic experiment |
| Component C: Atlantic gyre | C2.1 | dcppC-atl-spg | 3 | 200-400 | Predictability of 1990s warming of Atlantic gyre |
| | C2.2 | | 3 | 200-400 | Additional start dates |
| Component C: Volcano | C3.1 | dcppC-hindcast-noPinatubo | 1 | 50-100 | Repeat 1991 hindcast but without Pinatubo forcing |
| | C3.2 | dcppC-hindcast-noElChichon | 2 | 50-100 | Repeat 1982 hindcast but without El Chichon forcing |
| | C3.3 | dcppC-hindcast-noAgung | 2 | 50-100 | Repeat 1963 hindcast but without Agung forcing |
| | C3.4 | dcppC-forecast-addPinatubo | 1 | 50-100 | Repeat 2015 forecast with added Pinatubo forcing |
| | C3.5 | dcppC-forecast-addElChichon | 3 | 50-100 | Repeat 2015 forecast with added El Chichon forcing |
| | C3.6 | dcppC-forecast-addElChichon | 3 | 50-100 | Repeat 2015 forecast with added Agung forcing |

**Figures**

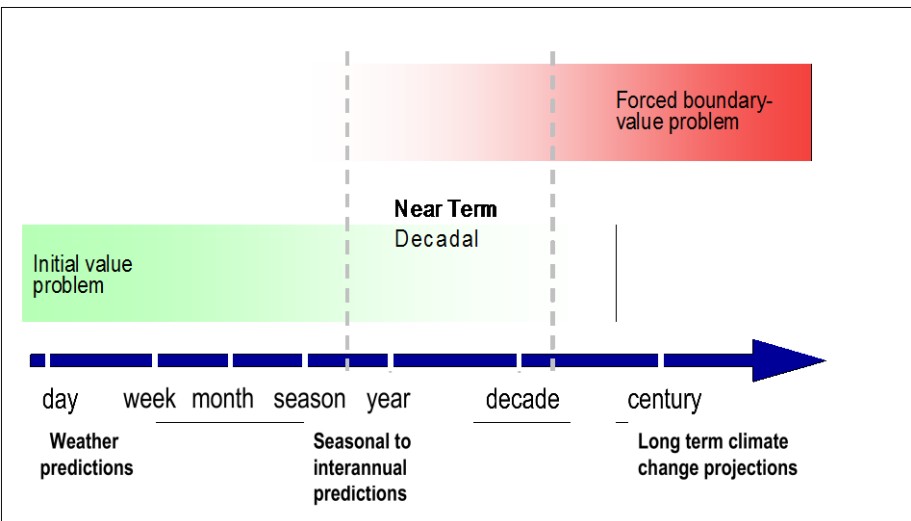

  **Figure 1.**   Predictions of interest to the Decadal Climate Prediction Project proceed from an initial condition problem at shorter timescales to a forced boundary-value problem at longer timescales (modified from Kirtman et al, 2013, Figure 11.2)

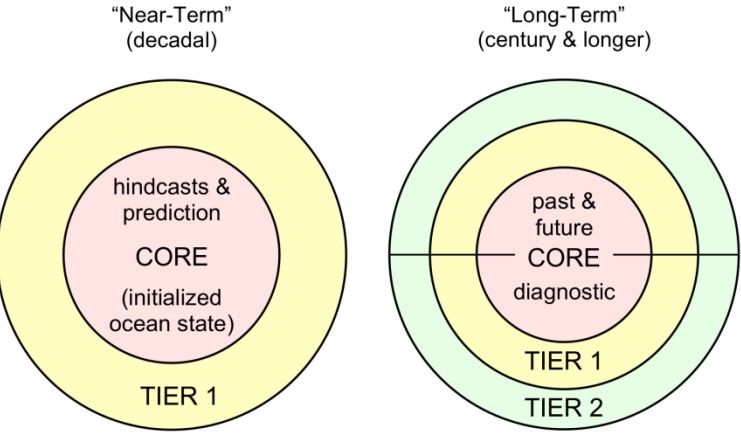

**Figure 2**.  Schematic of focus areas of CMIP5 divided into prioritized tiers of experiments (from Taylor et al., 2009). The DCPP structure is similar, but consists of three focus areas (Hindcasts, Forecasts, Mechanisms) each of which are tiered as summarized in Table 1 and in the Appendices as well as on the DCPP website.

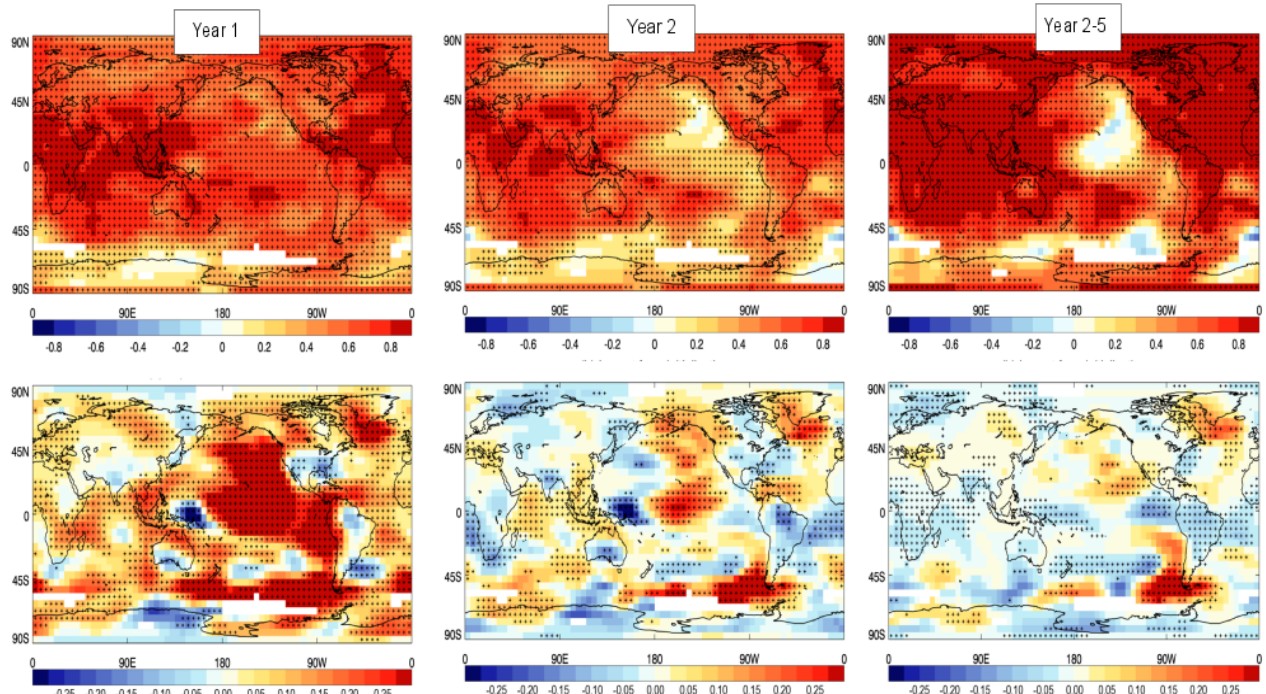

**Figure 3**. Correlation skill for Year 1, Year 2 and Year 2-5 forecasts of surface air temperature (upper panels). Impact of initialization (lower panels) based on the results from  CanCM4, GFDL,MPI-ESL-LR,MIROC5, HadCM3) and the HadCM3 PPE hindcsts. Stippling denotes that the results are significant at the 90% level (using a 2 tailed test). Plots provided by R. Eade (private communication).

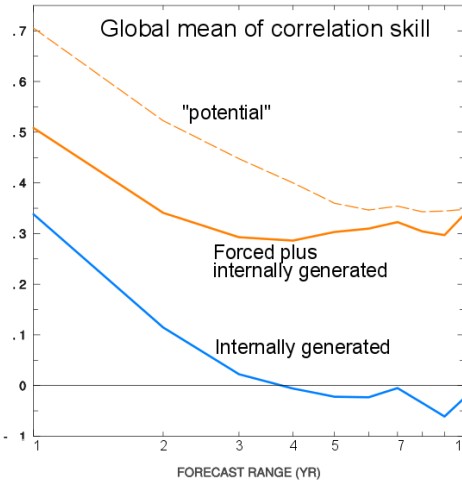

Figure 4. Global average correlation skill for surface air temperature from a single model (based on results from Boer et al., 2013). The orange curve plots the overall skill and the blue curve the skill associated with the initialized internally generated component. The difference between the two curves is associated with the forced component. The dashed line is an estimate of the "potential" skill that could be available if the actual system operated in the same fashion as the model.

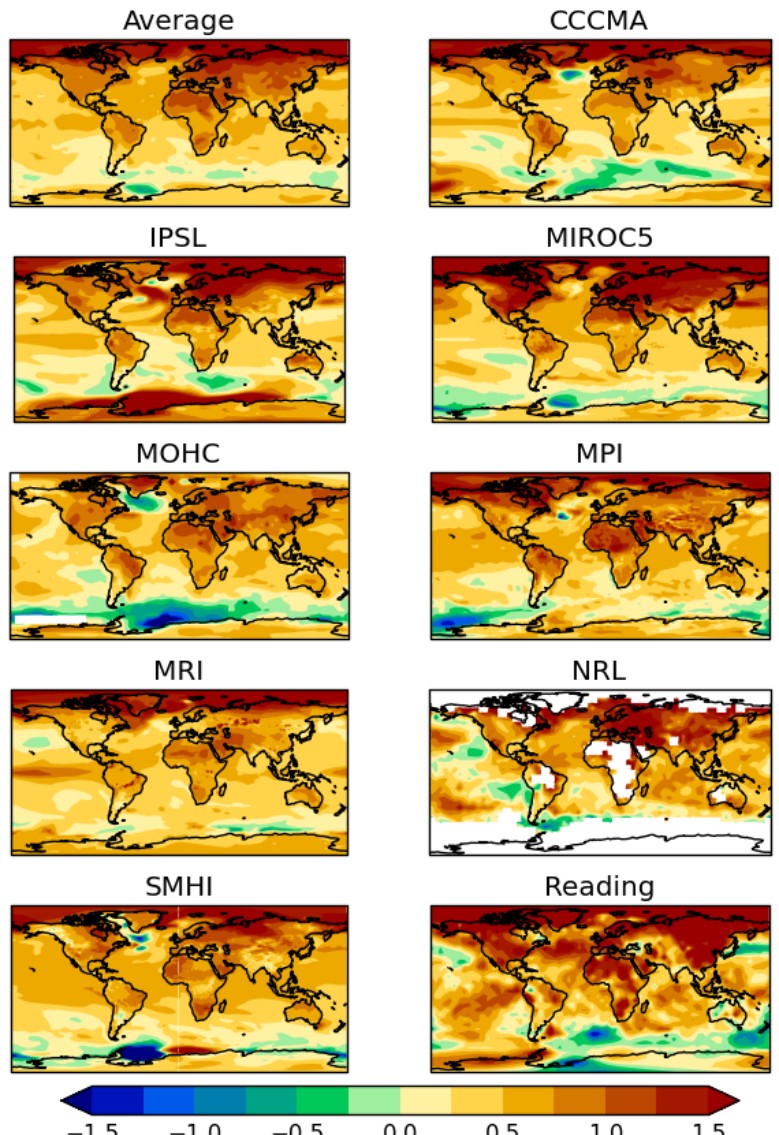

Figure 5. Example real-time multimodel decadal predictions (Smith et al. 2013, available from http://www.metoffice.gov.uk/research/climate/seasonal-to-decadal/long-range/decadal-multimodel). Maps show predicted near surface temperature anomalies ($^{o}$C) relative to the average over 1971 to 2000 for the 5 year period 2015-2019 from forecasts starting at the end of 2014.

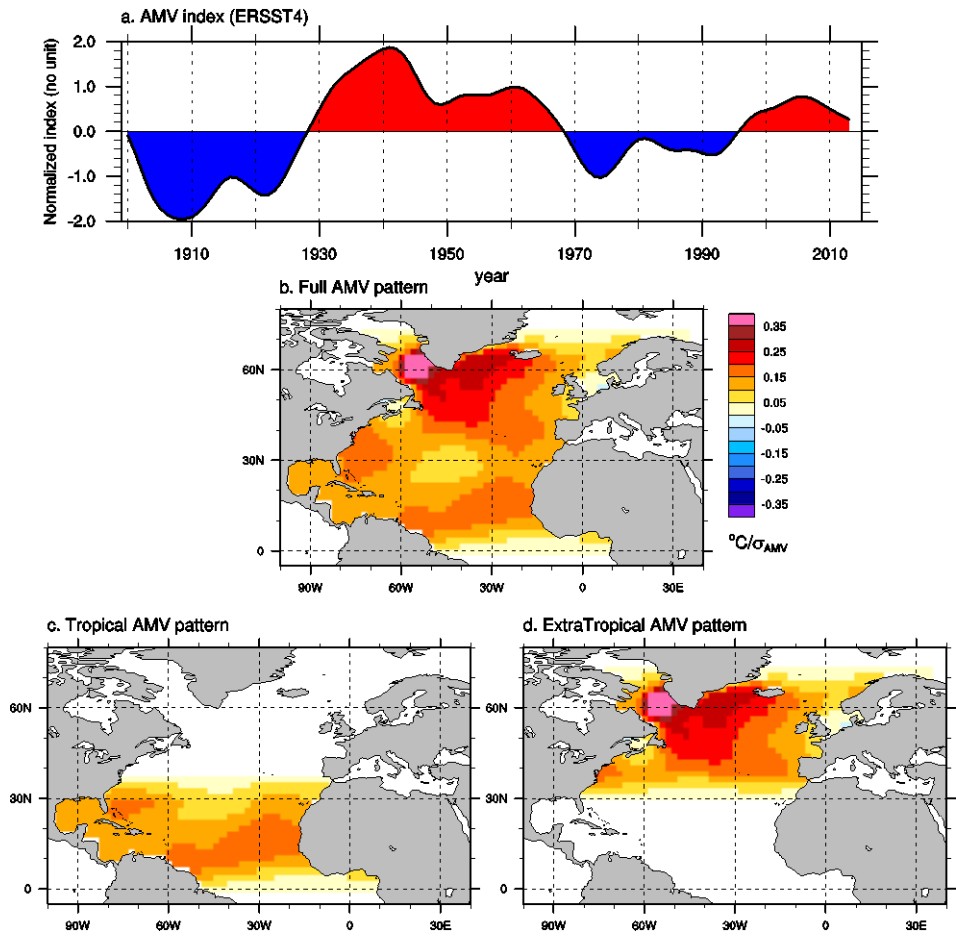

Figure 6. Idealized Atlantic SST patterns. The time series (upper panel) and pattern (middle panel) are derived following the procedure documented in Ting et al (2009) using ERSSTv4 (Huang et al. 2014) as discussed in Technical Note 1 (available from the DCPP website at http://www.wcrp-climate.org/dcp-overview). Experiments C1.1 to C1.3 use the total AMV pattern (middle panel), whereas experiments C1.7 and C1.8 apply anomalies in the northern extra-tropics and tropics separately (lower panels).

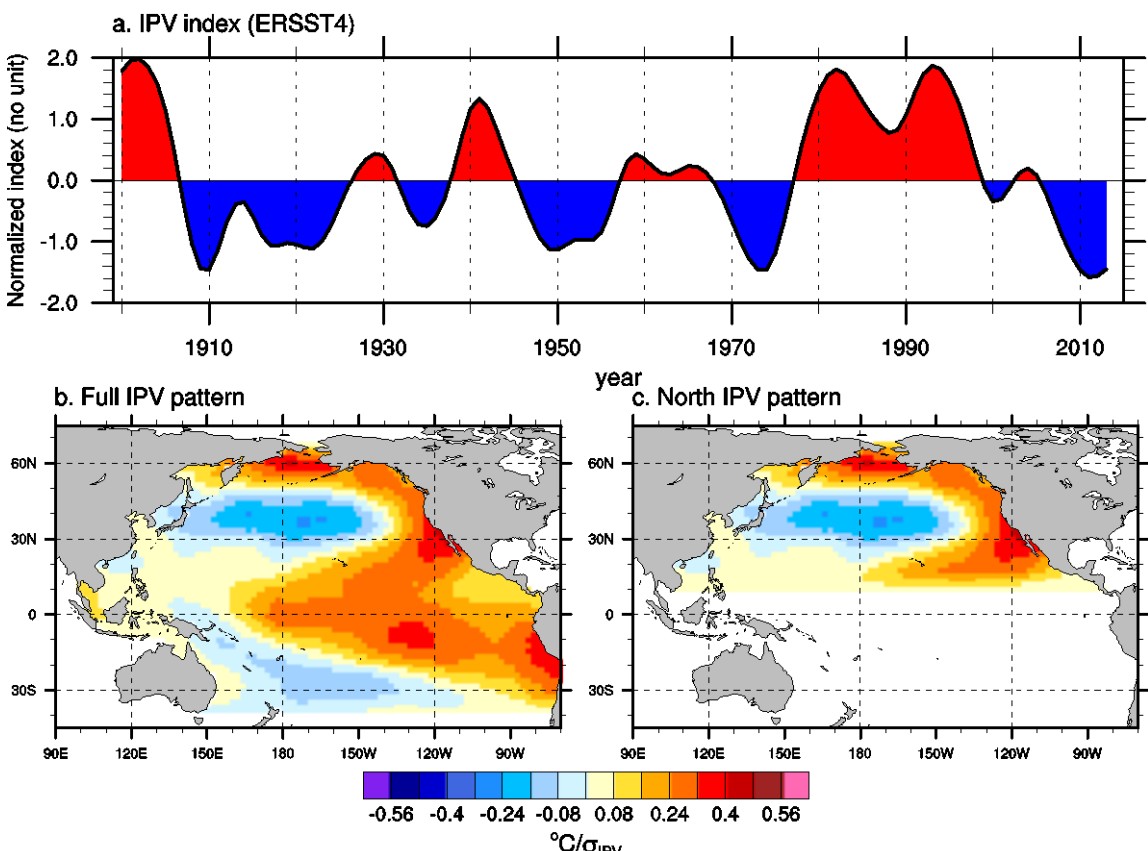

Figure 7. Idealized Pacific SST patterns. The time series (upper panel) and pattern (lower panel) are derived following the procedure documented in Ting et al (2009) using ERSSTv4 (Huang et al. 2014) as discussed in Technical Note 1. Experiments C1.4 to C1.6 use the full IPV pattern (lower left panel) whereas experiment C1.9 applies the anomalies in the northern extra-tropics (lower right panel).

### Apppendix A.  Component A hindcasts

The approach parallels that of the "Near-term Decadal" component of CMIP5 (Taylor et al. 2009) with important differences, notably that the hindcasts are to be produced every year, rather than every 5 years. As noted previously, "decadal" and "near-term" are used here to indicate annual, multi-annual up to ten year hindcasts. The Tier 1 experiment consists of hindcasts for years 1-5 for which the impact of initialization is expected to be greatest. Forecast skill is not geographically uniform and some regions will exhibit skill at longer timescales. The A2.1 experiment extends the hindcasts to years 6-10 to allow for the identification of these regions when resources permit. The A2.2 uninitialized historical simulations are compared with the initialized forecast to assess the impact of initialization.

**Table A1. Basic Component A: Hindcast/forecast experiments**

| # | Experiment | Notes | # of years |
|---|------------|-------|------------|
| **TIER 1: Hindcast/forecast information** | | | |
| A1 | Ensembles of 5 year *hindcasts* and *forecasts* | Coupled models with initialization based on observations<br><br>Start date *every year* from 1960 to the present (i.e. the first full hindcast year is 1961)<br><br>Start date on or before 31 Dec of the year preceding the forecast period (start dates on or before Nov 15 that allow for DJF seasonal forecast results are recommended)<br><br>10 ensemble members (more if possible)<br><br>Prescribed CMIP6 historical values of atmospheric composition and/or emissions (and other conditions including volcanic aerosols). Future forcing as the SSP2-4.5 scenario. | 60x10x5=3000 years of integration |
| **TIER 2: Increase the forecast range to 10 years** | | | |
| A2.1 | Extend the A1 hindcasts and forecasts | Extend the hindcasts and forecast in A1 for another 5 years up to and including year 10 | 60x10x5=3000 years of integration |
| **TIER 2: To quantify the effects of initialization (encompasses CMIP6/historical simulations)** | | | |
| A2.2 | Ensembles of historical and near-future climate *simulations* | Made with the same model as used for hindcasts<br><br>1850 to 2030, with initial conditions from a preindustrial control simulation<br><br>10 ensemble members (more if possible)<br><br>Prescribed historical and future forcing as for the A1 Experiment | 170x10=1700 years of integration |

**Table A2. Other hindcast experiments (if resources permit)**

| ▪ | Experiment | Notes | # of years |
|---|---|---|---|
| **TIER 3: Effects of increased ensemble size** | | | |
| A3.1 | Increased ensemble size for the A1 Experiment | $m$ additional ensemble members to improve skill and examine dependence of skill on ensemble size | 60x5x$m$=300$m$ years of integration |
| A3.2 | Increased ensemble size for the A2 Experiment | As A3.1 but for the A2.1 Experiment | 60x5x$m$=300$m$ years of integration |
| **TIER 4: Improved estimates of hindcast skill** | | | |
| A4.1 | Ensembles of at least 5-year, but much preferably 10-year, hindcasts and forecasts | As A1 but with no information from the future with respect to the forecast<br><br>Radiative and other forcing information (e.g., greenhouse gas concentrations, aerosols, etc.) maintained at initial state value or projected in a simple way. No inclusion of volcano or other short term forcing unless available at initial time. | 3000-6000 years of integration |
| **TIER 4: Improved estimates of the effects of initialization** | | | |
| A4.2 | Ensembles of at least 5-year, but much preferably 10-year, hindcasts and forecasts | Historical climate simulations up to the start dates of corresponding forecast with prescribed forcing<br><br>Simulations continued from forecast start date but with the same forcing as in A4.1, i.e. with NO forcing information from the future with respect to the start date. These are uninitialized versions of A4.1 hindcasts. | 3000-6000 years of integration |

**Table A1** lists the main DCPP Component A experiments. The A1 hindcast experiment parallels the corresponding CMIP5

5  decadal prediction experiment in using the same specified forcing as is used for the CMIP6 historical climate simulations. This forcing is also used for the historical simulations of experiment A2. For forecasts which extend beyond the period for which historical forcing is specified the "medium" SSP2-4.5 forcing of ScenarioMIP (described in a separate invited contribution this Special Issue) is used. This forcing scenario is used for several other MIPs and is chosen since "land use and aerosol pathways are not extreme relative to other SSPs (and therefore appear as central for the concerns of DAMIP and

10  DCPP), and also because it is relevant …. as a scenario that combines intermediate societal vulnerability with an intermediate forcing level". This forcing is also used for experiment A2.2 which is also a contribution to ScenarioMIP.

The specification of historical forcing introduces some information from the future with respect to the forecast and may lead to slightly overestimated historical forecast skill measures. The main effect is expected to be due to the specification of short term radiative forcings such as volcanoes which occur during a forecast. Other forcings, such as those associated with greenhouse gas and aerosol emissions and/or concentrations, vary comparatively slowly over the five or ten year period of a

5   forecast and are expected to have little effect on the results.  The benefits of using specified forcings include the use of common values across models, the ease of treatment within models, the possibility of documenting improvements with respect to CMIP5 hindcasts, the ability to estimate the effects of initialization by comparing forecasts and simulations which use the same forcings, and the estimation of drift corrections from hindcasts which include the forcings and so are more suitable for the purpose of future decadal forecasts.

Component A benefits from and builds on the experience gained from the decadal component of CMIP5. It calls for hindcasts every year, rather than every 5 years, which will improve the statistical stability of results, allow more sophisticated drift treatments, more clearly delineate skill levels, and foster improved assessment, combination, and calibration of the forecasts. Broad participation in Component A will potentially allow classification of results according to i)

the initialization of climate components in the models, ii) model resolutions including atmospheric model top, and iii) methods of initialization and ensemble generation. DCPP component A also provides an opportunity to study solar effects on climate. In order to take advantage of this, however, groups should use the correct ozone forcing time series which is important for the impact of solar variations.

**Table A2** lists additional experiments which are of interest if resources permit. The Tier 3 experiments, A3.1 and A3.2, increase the ensemble size in order to better isolate the predictable component in the case of a deterministic forecast and to better represent the probability distribution in the case of a probabilistic forecast. The A3 experiments may be used to help quantify the benefits of larger ensembles as a guide to future forecast applications. In the Tier 4 hindcasts the external forcing applied is based on information available at the start of the forecast (using persistence, extrapolation, or some other

method). This contrasts with the Tier 1 hindcasts where historical forcings are applied as discussed above It is not expected that many groups will undertake the Tier 4 experiments which require an additional large commitment of resources. They are included for completeness and in case the needed resources become available.

**Data retention.** See the CMIP6 website for links to the CMIP6 Data Request Compilation and CMIP6 Forcing Data Sets.
The DCPP input to the CMIP6 Data Request appears in Appendix D and applies to all experiment tiers.  Data are to be served via the ESGF and to parallel CMIP5 although with changes to protocols as specified by the WIP. At this time, 6-hourly decadal prediction data for dynamical downscaling are not considered a priority. The hope is that, in conjunction with the WIP, a coordinated set of "basic" or "common" tiered data tables can be developed across MIPs together with "MIP specific" tables associated with individual MIPs.

**Appendix B.**

**Component B:  Forecasts**

Objective:

- Production, collection and combination of real-time quasi-operational decadal forecasts

**Table B1. Real-time decadal forecasts**

| # | Experiment | Notes | # of years |
|---|---|---|---|
| **TIER 1: Real-time forecasts** | | | |
| B1 | Ensembles of ongoing real-time 5-year forecasts | Coupled models with initialization based on observations<br><br>Start date *every year* ongoing<br><br>Start date on or before 31 Dec (start dates on or before Nov 15 allow for DJF seasonal forecast results and are recommended)<br><br>10 ensemble members (more if possible)<br><br>Atmospheric composition and/or emissions (and other conditions including volcanic aerosols) to follow a prescribed forcing scenario as in A1. | 10x5=50 years of integration for 5-year forecasts |
| **TIER 2: Increased ensemble size and duration** | | | |
| B2.1 | Increase ensemble size | *m* additional ensemble members to reduce noise and improve skill | 5*m* years of integration |
| B2.2 | Extend forecast duration to 10 years | To provide forecast information for the period 5 to 10 years ahead | 10x5=50 years of  integration |

**Explanatory comment**

Component B real-time decadal forecasts are currently being produced based on CMIP5 and using other models and hindcast data sets. The intent is that the forecasts produced by these models will be augmented by Component A results as they become available**.** Data to be retained on the ESGF are the same as listed in the DCPP Data Retention Table in Appendix D. Data to be archived by January 31[st] of each year if possible.

# Appendix C.

## Component C: Predictability, mechanisms, and case studies

Component C consists of targeted simulations and prediction intend to: i) investigate the origins, mechanisms and predictability of long timescale variations in climate as well as their regional imprints and ii) to investigate the influence and consequences of volcanic eruptions on decadal prediction and predictability. See the Notes for details on methods and data.

## Component C1: Accelerated and retarded rates of global temperature change and associated regional climate variations

Objective:

- To investigate the role of Eastern and North Pacific and North Atlantic SSTs in the modulation of global surface temperature trends and in driving regional climate variations.

**Table C1.**

| # | TIER | Experiment | Notes | # of years |
|---|------|-----------|-------|-----------|
| **SST forcing experiments (see Notes and Appendices C1 and C2 for details)** | | | | |
| C1.1 | 1 | Idealised Atlantic control experiment | Restore North Atlantic SST to model control run climatology<br>-Time period: 10 years<br>- Region $10^o$N to $65^o$N (with $8^o$ buffer, see notes below)<br>-Ensemble size: 25 members, sampling different ocean states if possible<br>- Restoring of SSTs using a restoring coefficient of 40 $Wm^{-2}K^{-1}$, which is equivalent to about 2 months for a 50 m deep mixed layer<br>-No interannual changes in external forcings (set to pre-industrial control values)<br>- Minimization of drift if necessary | 25x10=250 years |
| C1.2 | 1 | Idealised climate impacts of AMV+ | As C1.1 but restore North Atlantic SSTs to positive AMV anomaly provided (Figure 6b) superimposed on model climatology | 25x10=250 years |
| C1.3 | 1 | Idealised climate impacts of AMV- | As C1.2 but for negative AMV anomaly pattern | 25x10=250 years |
| C1.4 | 1 | Idealised Pacific control experiment | As C1.1 but for the Pacific<br>- Region specified by PDV anomaly provided (Figure 7b)<br>- Ensemble size: 10 members | 10x10=100 years |

| C1.5 | 1 | Idealised climate impacts of PDV+ | As C1.4 but restore to positive PDV anomaly provided ( Figure 7b) superimposed on model climatology | 10x10=100 years |
|------|---|-----------------------------------|-------------------------------------------------------------------------------------------------------|-----------------|
| C1.6 | 1 | Idealised climate impacts of PDV- | As C1.5 but restore to negative PDV anomaly | 10x10=100 years |
| C1.7 | 2 | Idealised Atlantic extratropics | As C1.2 and C1.3 AMV+ and AMV- patterns but with restoring only in the extratropics( Figure 6c) Ensemble size: 25 members | 2x25x10=500 years |
| C1.8 | 2 | Idealised Atlantic tropics | As C1.7 but with restoring in the tropical band (Figure 6d) | 2x25x10=500 years |
| C1.9 | 2 | As C1.4 and C1.5 | As C1.4 and C1.5 but with restoring only in the northern extratropics (Figure 7c) | |
| C1.10 | 3 | Pacemaker Pacific: coupled model restored to observed anomalies of sea surface temperature in the tropical eastern Pacific | -Follow the experimental design of Kosaka and Xie (2013). -Time period: 1950 to 2014 (from 1910 if possible) -Ensemble size: 10 members or more. - Restoring timescales and ensemble generation as in C1.1 -Monthly SST anomalies (base period 1950-2014) are provided | 65x10=650 years |
| C1.11 | 3 | Pacemaker Atlantic: as above but for the North Atlantic | As C1.10 but restored to 12-month running mean SST anomalies (to be provided) in the North Atlantic, $10^{\circ}$N to $65^{\circ}$N -Time period: 1950 to 2014 -Ensemble size: 10 members (25 preferable) -Restoring timescales and ensemble generation: as for C1.1 -Minimization of drift if necessary (see Appendix C2) ) | 65x10=650 years |

**Component C2: Case study of mid-1990s Atlantic subpolar gyre warming**

Objectives:

- To investigate the predictability of the mid-1990s warming of the subpolar gyre and its impact on climate variability.

5   **Table C2.**

| # | TIER | Experiment | Notes | # of years |
|---|------|-----------|-------|-----------|
| | | | **Prediction experiments** | |
| C2.1 | 3 | Repeat hindcasts with altered initial conditions | Initialize with climatology (the average over 1960 to 2009) in the North Atlantic "sub-polar ocean"[95$^o$ W to 30$^o$ E, 45$^o$ N-90$^o$ N] <br> -Linear transition between climatology and actual observations over the 10$^o$ buffer zone 35$^o$ N-45$^o$ N <br> - 10 member ensembles <br> - 5, but much preferably 10 years <br> - Start dates end of 1993, 1994, 1995, 1996 | 4x(5,10)x10=200-400 years |
| C2.2 | 3 | Same as in C2.1 | As above with start dates 1992, 1997, 1998, 1999 | 200-400 years |

**Component C3: Volcano effects on decadal prediction**

Objectives:

- Assess the impact of volcanoes on decadal prediction skill
10   - Investigate the potential effects of a volcanic eruption on forecasts of the coming decade
- Investigate the sensitivity of volcanic response to the state of the climate system

**Table C3.**

| # | TIER | Experiment | Notes | # of years |
|---|------|-----------|-------|-----------|
| | | | **Prediction experiments with and without volcano forcing** | |
| C3.1 | 1 | Pinatubo | Repeat 1991 hindcasts without Pinatubo forcing <br> - 5 year, but preferably 10, year hindcasts <br> -10 ensemble members <br> -Specify the "background" volcanic aerosol to be the same as that used in the 2015 forecast | (5 or10)x10=50-100 years |
| C3.2 | 2 | El Chichon | 1982 hindcasts as above but without El Chichon forcing | 50-100 years |
| C3.3 | 2 | Agung | 1963 hindcasts as above but without Agung forcing | 50-100 years |

| # | TIER | Experiment | Notes | # of years |
|---|------|-----------|-------|-----------|
| **Prediction experiments for 2015 with added forcing** | | | | |
| C3.4 | 1 | Added forcing | Repeat 2015-2019/24 forecast with Pinatubo forcing | 50-100 years |
| C3.5 | 3 | Added forcing | Repeat 2015-2019/24 forecast with El Chichon forcing | 50-100 years |
| C3.6 | 3 | Added forcing | Repeat 2015-2019/24 forecast with Agung forcing | 50-100 years |

**Notes**

**Experiments C1.1-1.9** are idealized coupled model experiments following the methodology described in Ruprich-Robert et al. (2016) but with some changes. The AMV and IPV patterns used are displayed in Figures 6 and 7. The patterns are
derived from the difference between observations and the ensemble mean of coupled model historical simulations (Ting et al. 2009) and are an estimate of unforced internal variability. Although this estimate is not perfect because the modelled response to external factors such as anthropogenic aerosols may not be entirely correct, the experiments nevertheless provide information on the climate response to North Atlantic and Pacific SST variations. The experiments are based on model control integrations rather than historical simulations and therefore may be performed before the updated CMIP6 forcings
become available. See the DCPP website (http://www.wcrp-climate.org/dcp-overview) for links to Technical Note 1, which documents the methods used to produce the AMV and IPV patterns, and for links to the SST data to be used in the experiment.

**Experiments C1.10 and C11** follows the design of Kosaka and Xie (2013) in which observed SST anomalies are imposed
in the tropical Pacific region in coupled model simulations. The results will be compared to the standard historical simulations to infer the impact of the tropical Pacific SSTs.

These "pacemaker" experiments (C1.0 and C1.11) are of considerable interest in a multi-model context in which the response of the models to SSTs, imposed in the manner Kosaka and Xie (2013), is considered. Questions include the
robustness of the results across models, the geographic and global effects on climate and the pathways in the ocean and atmosphere through which the forcing is expressed. The experiments are Tier 3, however, because there may be coupled adjustment and drift issues that affect the results and this should be considered before undertaking the experiments. These include drift minimization (see below) and differences in variance and seasonality between models and observations. For these experiments:

- Observed monthly SST anomalies (base period 1950-2014) are superimposed on the model climatology over the same period computed from historical simulations in order to minimize model drift. ). See the DCPP website (http://www.wcrp-climate.org/dcp-overview) for links to this data)

    - Experiments should cover the period from 1950 to 2014, but starting from 1910 is desirable if possible.

- External forcings as for historical simulations.

**Methods of constraining SSTs and minimizing drifts are discussed in Technical Note 2**

    - The SST signal is imposed either by altering surface fluxes or by restoring the SST directly with no restoring if sea ice present. Outside of the restoring region, the model evolves freely allowing full climate system response.

- Experiments have shown that SST restoring, especially in the Atlantic, may lead to undesirable effects on ocean currents and associated heat transport such as AMOC which may affect SSTs in other regions (including the South Atlantic) and which can obscure the results. It is recommended that groups monitor this potential response and take steps to minimise it, if necessary following the recommendations in Technical Note 2 (available from the DCPP website at http://www.wcrp-climate.org/dcp-overview).

- In order to sample uncertainties in the ocean initial state it is recommended that, if possible, ensemble members are generated by taking initial conditions from different members of the historical simulations. Otherwise, ensembles may be generated by perturbing atmospheric conditions.

    - There is evidence that the signal to noise of the atmospheric response to North Atlantic SST is comparatively weak in models (Eade et al., 2014, Ruprich-Robert et al., 2016) and 25 ensemble members are requested, if possible. This contrasts

with the 10 ensemble members recommended for the Pacific experiments.

**Appendix D. DCPP Data Retention Tables**

The DCPP is concerned with prediction and a main interest is in variables that can be verified against observations.

Variables that provide insight into the ability to predict observed behaviour and the mechanisms involved are, of course, also of interest. There is a somewhat different emphasis on retained variables for the DCPP compared to the more usual approach which aims to study budgets, balances, processes etc. in the context of climate simulation rather than prediction. The large number of forecast years involved in the DCPP is also a consideration.

We stress that the DCPP data retention tables are *not intended to exclude* other variables. If modelling groups are willing and able to retain the variables requested by other MIPs also for the DCPP this would be ideal.

The following is intended as a prioritized set of variables for verification and investigation but is *not intended to restrict* the amount of data that groups retain for their DCPP integrations. With this understanding, the DCPP list is ordered into priorities as follows:

- Priority 1. These are basic forecast variables aimed at permitting bias adjusted forecast assessment, especially of well observed surface parameters and some atmospheric and oceanic structures, together with data that provides some information on the budgets and balances involved

- Priority 2. These are important variables that allow more detailed forecast assessment including, to some extent, predictions for the body of the atmosphere and ocean.

- Priority 3. These variables are intended for special interest investigations.

Participants should strive to retain at least Priority 1 variables and also Priority 2 variables to the extent that this is possible. Some basic discussion and recommendations on bias adjustment are given in Appendix E.

These tables are intended to provide an overview. Detailed specifications, including units etc., will be part of the "CMIP6 Data Request Compilation". The table headings indicate the nature of the data (e.g. TOA, BOA indicate top or bottom of the atmosphere) and the averaging period, yearly, monthly, daily or 6hour sampling. We have attempted to use standard CMIP5 variable names throughout although it is possible that there could be some differences with the CMIP6 Data Request Compilation. Three new variables, which lack CMIP5 standard names are indicated by an asterisk and names will be assigned by CMIP6.

**Table D. DCPP Data Retention Tables.**

| CMIP5 name | Short description | Averaging or sampling period and Priority | | | |
|---|---|---|---|---|---|
| | | Yr | Mon | Day | 6h |
| **TOA fluxes** | | | | | |
| rsdt | solar incident | | 1 | 3 | |
| rsut | solar out | | 1 | 3 | |
| rlut | lw out | | 1 | 3 | |
| rsutcs | clear sky solar out | | 2 | | |
| rlutcs | clear sky lw out | | 2 | | |
| **2D atmosphere and surface variables** | | | | | |
| tas | sfc air T | | 1 | 1 | 2 |
| tasmax | day T max | | 1 | 1 | |
| tasmin | day T min | | 1 | 1 | |
| uas | EW wind | | 1 | 2 | 2 |
| vas | NS wind | | 1 | 2 | 2 |
| sfcWind | day mean wind | | 1 | 1 | |

| | | | | | |
|---|---|---|---|---|---|
| sfcWindmax | day max wind | | 1 | 1 | |
| huss | specific humidity | | 1 | | |
| | | | | | |
| tdps | dewpoint temp | | 2 | 2 | |
| clt | cld frac | | 1 | 2 | |
| ps | sfc pres | | 2 | | |
| psl | mean sea level pressure | | 1 | 1 | 2 |
| | | | | | |
| Other high frequency data | | | | | |
| zg1000 | 1000hPa geopotential | | | | 2 |
| rv850* | 850hPa relative vorticity | | | | 3 |
| | | | | | |
| **BOA fluxes** | | | | | |
| rsds | solar down | | 1 | 1 | |
| rlds | LW down | | 3 | 3 | |
| rss | net solar | | 1 | 3 | |
| rls | net LW | | 1 | 2 | |
| tauu | EW stress down | | 2 | 3 | |
| tauv | NS stress down | | 2 | 3 | |
| hfss | sensible up | | 1 | 3 | |
| hfls | latent up | | 1 | 3 | |
| evspsbl | net evap | | 1 | | |
| pr | net pcp | | 1 | 1 | 2 |
| prsn | pcp as sno | | 3 | 3 | |
| prhmax | day pcp max | | 1 | 1 | 3 |
| **prmax** | **= prhmax?** | | | | 3 |
| | | | | | |
| **Land** | | | | | |
| *Physical variables* | | | | | |
| ts | skin temp | | 1 | | |
| | | | 1 | | |
| mrso | soil moist | | 1 | 3 | |
| mrfso | frozen soil moist | | 1 | | |
| snld | sno depth | | 1 | 3 | |
| **snd** | **= replace snld?** | | 1 | 3 | |
| mrro | runoff | | 1 | | |
| | | | | | |
| *Biogeophysical variables* | | | | | |
| treeFrac | tree fraction | 2 | | | |
| grassFrac | grass fraction | 2 | | | |
| shrubFrac | shrub fraction | 2 | | | |
| cropFrac | crop fraction | 2 | | | |
| vegFrac | total vegetated fraction | 2 | | | |
| baresoilFrac | bare soil fraction | 2 | | | |
| residualFrac | residual land fraction | 2 | | | |
| cVeg | vegetation carbon content | | 2 | | |
| cLitter | litter carbon content | | 2 | | |
| cSoil | soil carbon content | | 2 | | |

| | | | | |
|---|---|---|---|---|
| cProduct | carbon content of products of anthropogenic land use change | | 2 | |
| cLand | total land carbon | | 2 | |
| netAtmosLandCO2Flux | net atmosphere to land CO2 flux | | 2 | |
| gpp | gross primary productivity | | 2 | |
| npp | net primary productivity | | 2 | |
| lai | leaf area index | | 2 | |
| nbp | Surface net downward mass flux of CO2 as carbon due to all land processes | | 2 | |
| rh | heterotrophic respiration carbon flux | | 2 | |
| ra | plan respiration carbon flux | | 2 | |
| | | | | |
| **Sea Ice** | | | | |
| tsice | sfc temp | | 3 | 3 |
| sic | icefraction | | 1 | 3 |
| sit | ice thickness | | 1 | |
| snld | sno thickness | | 2 | 3 |
| hflsi | heat flux down | | 3 | |
| usi | EW ice speed | | 3 | |
| vsi | NS ice speed | | 3 | |
| strairx | EW stress down | | 3 | |
| strairy | NS stress down | | 3 | |
| | | | | |
| **2D Ocean** (preferably on regular grid) | | | | |
| *Physical variables* | | | | |
| tos | SST | | 1 | |
| sos | SSS | | 2 | |
| t20d | depth 20C | | 1 | 2 |
| mlotst | thickness mix layer | | 1 | 2 |
| thetaot | depth avg pot temp | | 1 | |
| thetao300* | depth avg pot temp to 300m | | 1 | |
| thetao700* | 700m | | 1 | |
| thetao2000* | 2000m | | 1 | |
| msftmyz | MOC | | 1 | |
| msftmyza | MOC atlantic | | 1 | |
| msftmyzba | bolus MOC **(msftmyzbasin?)** | | **2** | |
| hfnorth | northward ocean heat transport | | 2 | |
| hfnortha | Atlantic northward heat transport **(hfnorthbasin?)** | | **2** | |
| sltnorth | northward ocean salt transport | | 2 | |
| sltnortha | Atlantic northward salt transport **(sltnorthbasin?)** | | 2 | |
| zos | sea sfc height | | 1 | |
| zossq | square sea sfc height | | 2 | |
| zostoga | thermosteric sea level change | | 2 | |
| volo | volume sea water | | 2 | |
| hfds | net heat into ocean | | 1 | |
| vsf | virtual salt into ocean (or equivalent fresh water flux) | | 1 | |
| | | | | |

| | | | | | |
|---|---|---|---|---|---|
| *Biogeochemical variables(for ESMs)* | | | | | |
| epcalc100 | CaCO3 export @100m | | 2 | | |
| epsi100 | opal export @100m | | 2 | | |
| spco2 | surface aqueous partial pressure of CO2 | | 2 | | |
| fgco2 | surface downward CO2 flux | | 2 | | |
| co2s | atmospheric CO2 | | 2 | | |
| | | | | | |
| **3D Atmos (850, 500, 200, 100, 50)  Priority 1**<br>      **(925, 700, 300, 30,20, 10) Priority 2** | | | | | |
| ta | temp | | 1 | | |
| ta850 | temp 850 | | | 1 | |
| ua | EW wind | | 1 | | |
| va | NS wind | | 1 | | |
| hus | spec hum | | 2 | | |
| zg | geopotential | | 1 | | |
| zg500 | geopotential 500 | | | 1 | |
| wap | vertical press velocity | | 2 | | 2 |
| | | | | | |
| **3D Ocean** (preferably on a regular grid at standard levels) | | | | | |
| *Physical variables* | | | | | |
| thetao | pot temp | | 2 | | |
| so | salt | | 2 | | |
| uo | EW speed | | 2 | | |
| vo | NS speed | | 2 | | |
| wo | upward speed | | 3 | | |
| *Biogeophysical variables(for ESMs)* | | | | | |
| dissic | dissolved inorganic carbon concentration | | 2 | | |
| dissoc | dissolved organic carbon concentration | | 2 | | |
| talk | total alkalinity | | 2 | | |
| no3 | dissolved nitrate concentration | | 2 | | |
| o2 | dissolved oxygen concentration | | 2 | | |
| phyc | phytoplankton carbon concentration | | 2 | | |
| chl | total chlorophyll mass concentration | | 2 | | |
| zooc | zooplankton carbon concentration | | 2 | | |
| ph | seawater pH (reported on the total scale) | | 2 | | |
| pp | total primary (organic carbon) production by phytoplankton | | 2 | | |
| nh4 | dissolved ammonium concentration | | 2 | | |
| po4 | dissolved phosphate concentration | | 2 | | |
| dfe | dissolved iron concentration | | 2 | | |
| si | dissolved silicate concentration | | 2 | | |
| expc | sinking particulate organic carbon flux | | 2 | | |
| zfull | depth below geoid of ocean layer | 2 | | | |

These tables include some variables intend to aid the prediction/assessment/study of

- storm tracking
- energy production applications
- drought/flood studies
- sea level
- prediction of biophysical quantities

**Special Data Sets for consideration in support of other MIPs**

DCPP participants are encouraged to retain additional variables to support other MIPS, including DynVarMIP (Gerber and Manzini 2016), and to diagnose the effects of solar variability (Matthes et al 2016). Suggested variables include those for diagnosing the monthly mean momentum budget on pressure levels and monthly mean temperature and zonal winds on pressure levels up to and including 1 hPa.

**Appendix E: Bias correction for decadal climate predictions**

*Introduction*

No model is perfect and the result is a difference, or bias, between simulated and observed climatologies. This bias may introduce errors into a forecast that are large compared to the predictable signal. Here we update previous guidance (ICPO 2011) on how to correct biases in decadal predictions following discussions held at the SPECS/PREFACE/WCRP Workshop on Initial Shock, Drift, and Bias Adjustment in Climate Prediction (Barcelona, May 2016).

The two main approaches used to initialise forecasts for decadal predictions are full-field and anomaly initialisation. There is no clear advantage from either approach (Magnusson et al 2012, Hazeleger et al 2013, Smith et al 2013) and both are likely to be used in CMIP6.

In full-field initialization, models are initially close to the observations. However, as the forecast proceeds the model will drift towards its preferred climate state. The bias depends on the forecast lead time and its characterisation and correction requires a set of retrospective forecasts (also called hindcasts).

Anomaly initialization attempts to avoid drift by initializing models with observed anomalies (i.e. differences from the observed mean climate) added to the model mean climate obtained from historical simulations. Anomaly initialisation may, however, introduce dynamical imbalances leading to shocks and biases in the forecasts. Correcting for this source of bias also requires a set of hindcasts, and was not taken into account in ICPO 2011.

*Bias correction*

When comparing model simulations with observations, it is usual to consider anomalies from their respective means which corrects for differences in the means. For decadal forecasts the approach is further extended to the first order correction of the evolving bias. Assuming the bias is a function only of the forecast range it may be accounted for by calculating and comparing forecast and observation-based anomalies relative to their respective means at a particular forecast range. The same bias correction procedure is used for both full-field and anomaly initialisation.

Consider a set of "raw" climate forecasts $Y_{kj\tau}$ where $k$ denotes the ensemble member, $j$ identifies the initial times and $\tau$ is the forecast range. The observation-based information $X$ against which the forecasts are to be compared is labelled $X_{j\tau}$ to correspond with the forecasts. The mean of the observations at range $\tau$ is calculated as

$$\overline{X}_\tau = \sum_{j=year1}^{year2} X_{j\tau} / N_x$$

where $N_x$ is the *fixed* number of years with observations in the period *year1* to *year2* inclusive. The anomaly from the mean follows as

$$X'_{j\tau} = X_{j\tau} - \overline{X}_\tau$$

In like manner, the "forecast climatology" at range $\tau$ is calculated as

$$\{\overline{Y}\}_\tau = \sum_{j=year1}^{year2} \{Y\}_{j\tau} / N_y$$

where the ensemble mean forecast, obtained by averaging the ensemble members together, is denoted as $\{Y\}_{j\tau}$. Here, the average is over the $N_y$ forecasts that fall (but do not necessarily start) within the fixed *year1* to *year2* period. Anomalies follow as

$$Y'_{kj\tau} = Y_{kj\tau} - \{\overline{Y}\}_\tau$$

5   and observed and bias corrected forecast information is compared in terms of these anomalies.

It is important that the *year1* to *year2* period is the *same* for all forecast ranges in order to provide consistent estimates (Hawkins et al 2014) and avoid difficulties in interpreting forecasts relative to different baselines (Smith et al 2013). We recommend taking *year1* as 1970 and *year2* as 2016 for the DCPP Component A hindcasts that are part of CMIP6.

It is also important that the *year1* to *year2* period is as long as possible so as to sample multiple phases of variability and to provide robust estimates of the climatologies involved. Although we expect the number of hindcasts $N_y$ to equal the number of years $N_x$ between year1 and year2 there may be cases were forecast centers are unable to perform hindcasts starting every year. We nevertheless recommend using all $N_x$ observations in order to provide a robust estimate of the observation-based

15   climatology.

Finally, since forecast anomalies do not depend on observations, they may be calculated for unobserved or insufficiently observed variables (such as the Atlantic Meridional Overturning Circulation) although verification in these cases is not direct and is typically based on related observed variables.

*Derived quantities*

Some applications require derived quantities. Examples include (but are not limited to) predicting whether a particular threshold is likely to be exceeded, and predictions of storm frequency. In this case, the quantities of interest should be derived from the raw model data and then bias corrected as described above.

*Trends*

Differences in forecast and observed trends will not be corrected by the mean bias correction detailed above. In this case a bias correction that depends on the forecast start date may be necessary (e.g. Kharin et al 2012, Fuckar et al 2014). The most appropriate way to achieve this, especially for regional predictions, is a research question and several methods could be

30   considered (e.g. Gangstø et al 2013, Kruschke et al 2015).

*Further adjustments*

Many aspects of forecasts, in addition to the mean state and trends, may differ from the observed behaviour. There are a range of possible additional corrections but these are best considered separately from the basic correction considered here.

**Acknowledgements**

We are grateful to Environment and Climate Change Canada for providing publication support. Thanks to the Aspen Global Change Institute (AGCI) for hosting a DCPP workshop which contributed to Component C with funding from NASA, NSF,

40   NOAA, and DOE. DMS was supported by the joint DECC/Defra Met Office Hadley Centre Climate 503 Programme (GA01101) and the EU FP7 SPECS project. W. M Mueller was supported by the German Ministry of Education and Research (BMBF) under the MiKlip project (grant number 01LP1519A). NCAR is sponsored by the US National Science Foundation.

45

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
