# Peer review of "The Decadal Climate Prediction Project"

_Geoscientific Model Development, 2016_

## Short Comment (SC1) · 13 Apr 2016

Dear authors,

In agreement with the CMIP6 panel members, the Executive editors of GMD would like to establish a common naming convention for the titles of the CMIP6 experiment description papers.

The title of CMIP6 papers should include both the acronym of the MIP, and CMIP6, so that it is clear this is a CMIP6-Endorsed MIP.

Additionally, we strongly recommend to add a version number to the MIP description. The reason for the version numbers is so that the MIP protocol can be updated later, normally in a second short paper outlining the changes. See, for example: http://www.geosci-model-dev.net/special_issue11.html,

Good formats for the title include:

'XYZMIP (v1.0) contribution to CMIP6: Name of project'

or

'Name of Project (XYZMIP v1.0) contribution to CMIP6'

If you want to include a more descriptive title, the format could be along the lines of,

'XYZMIP (v1.0) contribution to CMIP6: Name of project - descriptive title'

or

'Name of Project (XYZMIP v1.0) contribution to CMIP6: descriptive title.'

When you revise your manuscript, please correct the title of your manuscript accordingly.

Yours,

Astrid Kerkweg

———————————————

---

## Referee Comment (RC1) · Anonymous Referee #1 · 9 May 2016

This manuscript sets out the protocol for the forthcoming Coupled Model Intercomparison Project (CMIP) 6 experiments on initialised climate prediction. It is only the second time that such experiments were included in CMIP and the current manuscript sets out necessary experimental plans as well as providing an iteration towards an improved set of hindcast experiments compared to CMIP5. The contents are the result of numerous international meetings and discussions and represent the synthesis of a large body of expert scientific opinion. I suggest this important protocol is published after minor technical corrections.

p1 line 25: please add a reference to Eyring et al 2015 at this early stage when CMIP6 is first mentioned

p2 line 10: the GC on Near Term Prediction under WCRP is now approved so please update this statement

p2 line 33: evolution

p3 line 6: ...their individual contribution...

p3 line 10: ...opertional climate predictions on annual....

p3 line 24: suggest a reference to Smith et al, ERL, 2012 regarding the potential for improved predictions

p3 line 25-27: I think it is also important to note that enhance skill arises from the longer averaging periods (e.g. yrs 1-5) normally adopted in this area

p5 line 18: please add a couple of references to the idea of continued improvement. Bauer et al Nature 2015 and MacLachlan et al QJRMS 2015 give examples for weather and seasonal forecasting respectively

p7 line 8-9: suggest alternate wording: "...assessment of performance. Skilful real time mulitannual forecasts will be a contribution to the GFCS and fill the gap between seasonal predictions and long term climate projections."

p7 line 22: ...and applications communities includind National Meteorological and Hydrological Services and Regional Climate Centres.

p8 line 7: explain DCVP on first use

p8 line 27: there is a question mark after 7

p10 line 6: perhaps it would be wroth referenceing the WMO pbook on this topic "Climate Science for Serving Society" by Asrar and Hurrell (Eds)

p11: Just for information: is it worth saying why C1.9, C1.10 are longer than other experiments?

Fig.2 caption: ...summarized in Table 1...

Fig.3 caption: which models are used?

Table A2: I am unclear as to why 1500-6000y of simulation are required for A4.1 and A4.2. Is this not the same as A1?

Table A2: I find the notes here very confusing please clarify what this experiment is for and also what is meant by "continued from forecast start date"

Table A1 caption: double space before "the" in line 2

p26 Line 33: Eade et al GRL 2014 showed weak Atlantic response in initialised predictions and precedes the reference here so please add.

---

## Short Comment (SC2) · 6 Jun 2016

**Comments from CMIP Panel**

The CMIP Panel is undertaking a review of the CMIP6 GMD special issue papers to ensure a level of consistency among the invited contributions, also in answering the key questions that were outlined in our request to submit a paper to all co-chairs of CMIP6-Endorsed MIPs. These questions are outline in the overview paper (Eyring et al, GMD, 2016) and the relevant section is summarised below:

*'Each of the 21 CMIP6-Endorsed MIPs is described in a separate invited contribution to this Special Issue. These contributions will detail the goal of the MIP and the major scientific gaps the MIP is addressing, and will specify what is new compared to CMIP5 and previous CMIP phases. The contributions will include a description of the experimental*

[Figure]

*design and scientific justification of each of the experiments for Tier 1 (and possibly beyond), and will link the experiments and analysis to the DECK and CMIP6 historical simulations. They will additionally include an analysis plan to fully justify the resources used to produce the various requested variables, and if the analysis plan is to compare model results to observations, the contribution will highlight possible model diagnostics and performance metrics specifying whether the comparison entails any particular requirement for the simulations or outputs (e.g. the use of observational simulators). In addition, possible observations and reanalysis products for model evaluation are discussed and the MIPs are encouraged to help facilitate their use by contributing them to the obs4MIPs/ana4MIPs archives at the ESGF (see Section 3.3). In some MIPs additional forcings beyond those used in the DECK and CMIP6 historical simulations are required, and these are described in the respective contribution as well.'*

We very much welcome the important contribution from DCPP to the CMIP6 special issue. We think the experiment design is well chosen and reflects the long and intense community discussions that DCPP has initiated during the last years, but we have several comments on the manuscript that we are hoping you can address in a revised version:

Please ensure that the title of your paper includes both the acronym of the MIP, and CMIP6, so that it is clear this is a CMIP6-Endorsed MIP (e.g.*'The Decadal Climate Prediction Project (DCPP) contribution to CMIP6'*).

Please ensure consistency of the experiment names with the CMIP6 overview paper (see Eyring et al., 2016).

p1, l5ff The state-of-the art on decadal predictions that has been achieved with CMIP5 simulations and a proper definition or motivation for decadal predictions seems missing from the introduction and the rest of the paper. Related to this then it is difficult to see how this new effort for CMIP6 builds on achievements from CMIP5 and what is different or better. For example, are Component A Tier 1 experiments a simple rep-

etition of CMIP5 hindcasts or are they different in terms of e.g. new observations for initialization, new initialization techniques, improved models, or initialized additionally with non-ocean components (e.g. sea-ice, land, stratosphere)? The sentence p4, l16 *'The lessons learned from the CMIP5 decadal prediction experiments have been incorporated into the design of the DCPP.'* is a good start, but the lessons should actually be summarized in the paper.

The CMIP6-Endorsed MIP contributions to the GMD special issue have been asked to describe the experimental design of each of the experiments in detail, so that the groups can find all the necessary information how to run the experiments in each paper, with technical instructions that could be given in an appendix. To achieve this, we feel that additional information is required that could be provided in the appendices. For example. (1) There is no mentioning or recommendation regarding full-field initialization versus anomaly initialization; (2) we assume the forcings in Component A and B should be exactly as in the CMIP6 historical simulations or a specific ScenarioMIP experiment. However, this is not mentioned. Please expand. Even if in some cases the authors decide not to provide a specific recommendation for good scientific reasons, then this is fine but should be explained and justified.

The scientific analysis plan is rather general and we are hoping it can be extended. All the manuscript currently says is summarized in a single sentence without references: *'The analysis of available observations for initializing forecasts, the improvement of the models used in the production of the forecasts, post processing of forecasts including bias adjustment, calibration and multi-model combination, together with the production and application of probabilistic decadal forecasts, are all involved in the research and development efforts contributing to the DCPP.'* There are a lot of open questions in the analysis of decadal simulations (e.g., bias corrections, different metrics used to assess skill) which could benefit from some guidance from DCPP. Even if DCPP is not able to recommend something specific, a scientific summary with references to some of the key papers should be given.

Figure 3: Please expand the caption and text so that it is clear what the figure shows.

For each proposed experiment to be included in CMIP6, the paper should specify the science question and/or gap being addressed with this experiment and should outline possible synergies with other CMIP6-Endorsed MIPs. (1) There is no mention of the other CMIP6-Endorsed MIPs in the paper, but there are some obvious relationships for example with ScenarioMIP for the forecast component (Component B), to DAMIP (for the additional ensemble members of the CMIP6 historical simulation), to GMMIP that also defines pacemaker experiments etc. Please expand this discussion in the main paper. (2) The scientific motivation for the experiments in the main paper is summarized only very briefly in a few bullet points collectively for each of the three components. We suggest that this list is expanded so that the scientific motivation is provided in the main part of the manuscript. For example, there are good reasons why a larger ensemble size is also required for the CMIP6 historical simulations in order to properly assess hindcast skill. Or, while those involved in the discussions know why an experiment like C3.1 is suggested, the scientific motivation is not provided in the manuscript, leaving the unfamiliar reader with speculation why such an experiment might possibly be useful. Please expand.

p19, l20ff: Demonstrated connectivity to the DECK and the CMIP6 historical simulations is an endorsement criterion for the MIPs (see No 2 in Table 1 of Eyring et al., 2016). We suggest to move this discussion from the Appendix to the main text. Please convincingly state how the models that do not run DECK/Historical are otherwise sufficiently characterized and how drifts, unforced variability and climate sensitivity that matter for predictions on decadal time-scales are diagnosed or why it is believed this does not matter.

Component A is a basis for forecasting on annual to decadal timescales as is said in the abstract. Therefore it seems important that Component A is performed by all models that enter Component B of DCPP. We suggest to rewrite the last sentence of the abstract to reflect this: *'Groups are invited to participate in as many or as few of the*

*Components of the DCPP, each of which are separately prioritized, as are of interest to them.'* This should then be stated again in the main manuscript.

p.4,l9: *'although both climate simulations and decadal hindcasts'*: decadal hindcasts are also climate simulations. Please be more specific.

p.10,l.8 ff: The last sentences of the data availability section gives the impression that separate forcings are developed for DCPP. However, hindcasts and forecasts use exactly the same forcings than the CMIP6 historical simulations and ScenarioMIP, respectively. Please rewrite this part accordingly, for example *'Datasets of natural and anthropogenic forcing information are required for the DCPP simulations, which are based on the exact same forcings that are defined for the CMIP6 historical simulations and ScenarioMIP. They are described in separate contributions to this special issue and will be made available through the ESGF with version control and DOIs assigned.'*

p.26, l13: The pacemaker experiments in Component C refer to the design of Kosala and Xie (2013). Please specify the experiment to ensure common treatment among the models (e.g. in terms of handling of sea-ice and what exactly is prescribed (wind stress / SSTs)). We would like to encourage you to liaise with GMMIP on the pacemaker experiments, since GMMIP is defining similar experiments and the methods used should be the same or explanations for differences should be given.

Table A1. The entry says *'Prescribed CMIP6 historical values of atmospheric composition and/or emissions and other conditions including volcanic aerosols. Future forcing as the SSP2-4.5 scenario.'* We believe this entry should say: *'Forcings in the hindcasts exactly as in the CMIP6 historical simulations and for forecasts as in the SSP2-4.5 scenario of ScenarioMIP'.* Please give a reference for SSP2-4.5 and define the acronym. Why this scenario and not another one? Please motivate in the main text.

Table A1, experiment A2.2: why are the ensemble members from the CMIP6 historical simulations requested from 1850 onwards? For the particular analysis of DCPP,

[Figure]

wouldn't it be sufficient to ask for ensembles starting in 1960? Please motivate.

Reference:

Eyring, V., Bony, S., Meehl, G. A., Senior, C. A., Stevens, B., Stouffer, R. J., and Taylor, K. E.: Overview of the Coupled Model Intercomparison Project Phase 6 (CMIP6) experimental design and organization, Geosci. Model Dev., 9, 1937-1958, doi:10.5194/gmd-9-1937-2016, 2016.

With many thanks for your ongoing efforts in the CMIP6 process.

The CMIP Panel

---

## Referee Comment (RC2) · N. Keenlyside (Referee) · 4 Jul 2016

This is a clear and well motivated description of experiments contributing to the DCPP. I have only minor comments, which I think may improve the text (descriptions and motivation) a little.

1. Pg 3, L25, fig. 1, It is difficult to argue that initialisation has enhanced prediction skill in years 2 and beyond, as there seems to be equal or greater areas of negative skill. I suggest a more careful formulation, supported by reference to process understanding. In particular, the skill in the NA sub polar gyre has been attributed to initialisation.

2. Pg 5, L15, I undertand that by "analysis" is used here to refer to data assimilation. This terminology may not be obvious to many readers. Also the list of contributors misses "enhancement of the observing system" which could be argued to be the most important. Furthermore, you could include statistical (flux correction, and anomaly coupling, and anomaly initialisation) methods that reduce forecast drift.

3. Pg 6, I believe that developing an understanding of the impact of initial shock on forecast skill should be mentioned under scientific aspects (perhaps under point 2).

4. Pg 17, Appendix A and respective place in main text, I can partly understand the reasoning for limiting the tier-1 hindcasts experiments to years 1-5, however, I would call them mulitannual and not near-term or decadal. Personally, I feel the greatest benefit comes from the longer 1-10 year period that focus on capturing predictability associated with the low-frequency component of climate variability rather than the interannual that is dominated by ENSO (which is not predictable beyond a year). It could be useful to make clear why shorter hindcasts are being encouraged and also called "near-term", which I understand refers to 10-30 years periods.

5. Appendix C. Is there any shock expected from applying a temperature anomaly essentially instantaneously in experiments C1.1-C1.8? If there is one, it could introduce an artefact into the results. How will it be assessed? For experiments C1.9 and C1.10, is there a reason for suggesting to start the extended pacemaker experiments exactly in 1920. The early century warming started in 1920, and it wouldn't be prudent to start the runs a little earlier if this is of interest.

6. Pg 29, I think it is important to also include salinity data (surface, upper 300m, 700m and 2000m) in the 2D Ocean data. These quantities are important for verifying the mechanisms for multi-decadal variability in the North Atlantic.

Typos
1. Pg8, L24, It should be: "to what extent can"
2. Pg 17, Table 17. A1, It should be "and start ….are recommended"
3. C1.1, I believe you mean a 50m deep mixed layer.

---

## Author Comment (AC1) · 8 Aug 2016

We have adopted the recommendation and have renamed the paper "The Decadal Climate Prediction Project (DCPP) contribution to CMIP6". However, in keeping with the wishes of the CMIP6 Panel we have avoided version numbers.

---

## Author Comment (AC2) · 10 Aug 2016

**The Decadal Climate Prediction Project**
**Responses to Anonymous Referee #1**

We appreciate very much the referee's positive and helpful comments and accept essentially all of his suggestions in the revised version of the paper as noted below.

All of the comments below are accepted and changes made as suggested.
- p1 line 25: please add a reference to Eyring et al 2015 at this early stage when CMIP6 is first mentioned
- p2 line 10: the GC on Near Term Prediction under WCRP is now approved so please update this statement
- p3 line 6: ...their individual contribution...
- p3 line 10: ...opertional climate predictions on annual....
- p3 line 24: suggest a reference to Smith et al, ERL, 2012 regarding the potential for improved predictions
- p3 line 25-27: I think it is also important to note that enhance skill arises from the longer averaging periods (e.g. yrs 1-5) normally adopted in this area
- p5 line 18: please add a couple of references to the idea of continued improvement Bauer et al Nature 2015 and MacLachlan et al QJRMS 2015 give examples for weather and seasonal forecasting respectively
- p7 line 8-9: suggest alternate wording: "...assessment of performance. Skilful real time mulitannual forecasts will be a contribution to the GFCS and fill the gap between seasonal predictions and long term climate projections."
- p7 line 22: ...and applications communities includind National Meteorological and Hydrological Services and Regional Climate Centres.
- p8 line 7: explain DCVP on first use
- p10 line 6: perhaps it would be wroth referenceing the WMO pbook on this topic "Climate Science for Serving Society" by Asrar and Hurrell (Eds)
- Fig.2 caption: ...summarized in Table 1...
- Table A1 caption: double space before "the" in line 2
- p26 Line 33: Eade et al GRL 2014 showed weak Atlantic response in initialised predictions and precedes the reference here so please add.

Other changes that have been made in response to comments
- p8 line 27: there is a question mark after 7
  - this is a typo, now corrected
- p11: Just for information: is it worth saying why C1.9, C1.10 are longer than other experiments?
  - the experiments that involve the AMV and IPV are 10 year experiments and since the response to AMV appears to be weaker than to IPV larger ensembles are suggested to help overcome this and this is now mentioned more explicitly in the notes
  - the pacemaker experiments referred to differ in the length of the simulations involved, 65 years vs 10 years, and are different in implementation as compared to  the AMV and IPV experiments leading to the different total number of years

- Fig.3 caption: which models are used?
  - these are now listed in the text and referred to in the caption
- Table A2: I am unclear as to why 1500-6000y of simulation are required for A4.1 and A4.2. Is this not the same as A1?
  - yes, this was a typo and is now corrected
- Table A2: I find the notes here very confusing please clarify what this experiment is for and also what is meant by "continued from forecast start date"
  - this has been rewritten in what we hope is a clearer manner

---

## Author Comment (AC3) · 10 Aug 2016

**The Decadal Climate Prediction Project**
**Responses to Comments SC2**

We appreciate the time and effort that the CMIP6 Panel has put in to review the description of the DCPP experiment. It is always helpful to receive comments that indicate how the paper is read and perceived by other than the authors who are close to the material. The other reviewers were basically satisfied with the organization and the "style", if we may call it that, of the paper where we have attempted to be reasonably terse and to concentrate on the specification of the coordinated experiments that form the DCPP contribution to CMIP6.

We have purposefully avoided writing a review of decadal prediction results and have instead referred to a few basic and recent publications which also provide lists of pertinent references. We provide some additional references but do not attempt the many references which would be needed to cover the many and very broad aspects of the DCPP. We have assumed that participants in DCPP/CMIP6 will understand the basic scientific context and so have been comparatively terse in this regard also. Our attempt is to write the paper as a reference for potential participants who will undertake some or all of the experiments proposed. We do our best to respond to the CMIP6 Panel comments below.

1. Please ensure that the title of your paper…. contribution to CMIP6').

We have adopted this very title. We have also, in compliance with the wishes of the Panel, avoided a version number.

2. p1, l5ff  The state-of-the art on decadal predictions that has been  achieved with CMIP5……Please expand.
      We respond to what we consider are the several aspects of the comments:
        i.    what does CMIP6 bring beyond CMIP5
       ii.    what level of information is needed to specify the experiments (e.g full field or anomaly initialization) etc.?
     iii.    forcing should be specified more clearly
     iv.    guidance on the analysis of results
      v.    Figure 3
     vi.    further description of science question/gaps, motivation (e.g. ensemble size)
    vii.    connections with other MIPs

Responses:

2iii.  Please see the responses to 6 and  8 below concerning forcing.

2v. Figure 3. We do not understand what is missing in the caption of Figure 3 which is a more or less standard in decadal prediction result. We have added the identifiers of the models involved to the caption and modified the text to be more explicit.

2i,ii,iv,vi. We have considerably rewritten and expanded the section beginning p5,l5 in an attempt to respond to these comments but without attempting a review of the very broad range of material

involved. We fairly often refer to the recent IPCC report as a heavily referenced compendium of recent published material and we also add some other references. The hope is that this expanded material provides at least some of the information that is felt to be missing. In particular we have added subsections "Multi-system approach", "Analysis of results", "Deck and CMIP6 historical simulations" and "Participation".

The expanded sections provide some further discussion of the "science questions/gaps" which we hope are helpful. As noted earlier, our intent is to provide some terse background while concentrating on the specification of the coordinated experiments that form the DCPP contribution to CMIP6. We do not approach the GMD paper as scientific motivation for an unfamiliar reader but do add some text concerning the desirability of larger ensembles for instance. We have kept the motivation brief in the body of the paper and added details in the appendices for the more motivated reader.

As also noted in the expanded text, we make no recommendations as to the details of initialization for instance. We have adopted the view that the DCPP prescribes a specific experimental design but not the details of the implementation. This, of course, is entirely in the tradition of past CMIP approaches to both simulation and prediction. We do not recommend model resolutions; physical parameterizations, specific methods of initialization etc. etc. since the evidence for the best approach is not available and will, in part, be revealed by the output of the DCPP. The presumption is that the participants will naturally adopt what they regard as the best approaches based on their understanding of their forecasting systems and that this is suitable input to a "multi-system" approach.

2.vii. A brief section noting DCPP connections with ScenarioMIP, DAMIP, VolMIP, DynVar and SolarMIP has been added although details of the connections are not stressed. Recent interactions with GMMIP have resulted in common specification of some experiments which formerly differed from those of the DCPP.

3. p19, l20 Demonstrated connectivity ….does not matter.
See response above and the new subsection "Deck and CMIP6 historical simulations"

4. Component A ...
Component A. It is correct that Component A results will (as noted in the expanded text) support the results of Component B. However, we do not insist that only models that have completed Component A can submit results to Component B. Component B may be willing to consider results that are based on other hindcast data sets, especially in the interim, while Component A results are being generated.

5. p4, l9 …
We don't agree that decadal hindcasts are also climate simulations. Common usage (e.g. Chapter 11, IPCC 2013) note that while simulations represent possible evolutions of the system under external forcing and independent of initial conditions, predictions attempt to trace out the actual evolution of the system based on the initial state plus the external forcing.

6. p10,l8…"forcing"
Yes we agree that this is important and have adopted this text in Section 12 Data Availability.

7. p26 l13…
As seen at l20 the imposed tropical SSTs for the pacemaker experiments are made available on the PCMDI website. There are no references to winds or wind stress, which are not part of the experiment and the treatment where sea ice exists is specified.

8. Table A1. We now reference ScenarioMIP and motivate the choice of SSP2-4.5 as characterized there. Historical simulations, as such, cannot start from 1960 since they depend on past forcing so the entire period is involved.  The retention of data from 1850 contributes to the CMIP6 historical multi-model ensemble.

---

## Author Comment (AC4) · 10 Aug 2016

**The Decadal Climate Prediction Project**
**Responses to Referee #2**

1. Pg 3, L25, fig. 1, It is difficult to argue that initialisation has enhanced prediction skill in years 2 and beyond, as there seems to be equal or greater areas of negative skill. I suggest a more careful formulation, supported by reference to process understanding. In particular, the skill in the NA sub polar gyre has been attributed to initialisation.

- Sorry for the typo where Fig 1 should read Fig3. We have modified the text in response to this comment and made reference to the North Atlantic and Component C.

2. Pg 5, L15, I undertand that by "analysis" is used here to refer to data assimilation. This terminology may not be obvious to many readers. Also the list of contributors misses "enhancement of the observing system" which could be argued to be the most important. Furthermore, you could include statistical (flux correction, and anomaly coupling, and anomaly initialisation) methods that reduce forecast drift.

- We have added text with respect to initialization, ensemble generation, and the coupling of model components as suggested. Although we expect enhancement of the observing system to improve initial conditions and hence prediction skill we don't feel we can appeal to it in the decadal context as yet. Presumably ARGO and other enhancements will do this but we don't think this has been demonstrated so far.

3. Pg 6, I believe that developing an understanding of the impact of initial shock on forecast skill should be mentioned under scientific aspects (perhaps under point 2).

- We don't disagree that this is important but would like to leave it  as an implicit topic under the heading of "broad questions" rather than  mentioning it, and other specific points, in this section

4. Pg 17, Appendix A and respective place in main text, I can partly understand the reasoning for limiting the tier-1 hindcasts experiments to years 1-5, however, I would call them mulitannual and not near-term or decadal. Personally, I feel the greatest benefit comes from the longer 1-10 year period that focus on capturing predictability associated with the low-frequency component of climate variability rather than the interannual that is dominated by ENSO (which is not predictable beyond a year). It could be useful to make clear why shorter hindcasts are being encouraged and also called "near-term", which I understand refers to 10-30 years periods.

- We have added text intended to clarify the usage of "decadal" and "near term" and have also added text to indicate that the longer timescale predictions are both important and encouraged when resources permit.

- Unfortunately the terminology is a bit vague in this area and "near-term" is used to  mean 10-30 years in Chapter 11 of the IPCC for instance but 1-10 years in the WCRP Grand Challenge of Near Term Climate Prediction. We follow this latter usage.

5. Appendix C. Is there any shock expected from applying a temperature anomaly essentially instantaneously in experiments C1.1-C1.8? If there is one, it could introduce an artefact into the results. How will it be assessed?

- Technical Notes which discusses methods of imposing the temperature anomalies and for minimizing potential shock and drift are now available.

For experiments C1.9 and C1.10, is there a reason for suggesting to start the extended pacemaker experiments exactly in 1920. The early century warming started in 1920, and it wouldn't be prudent to start the runs a little earlier if this is of interest.

- Agreed, we now suggest 1910

6. Pg 29, I think it is important to also include salinity data (surface, upper 300m, 700m and 2000m) in the 2D Ocean data. These quantities are important for verifying the mechanisms for multi-decadal variability in the North Atlantic.

- Surface salinity has been added (it was left out by accident) and salinity data is requested under 3D ocean variables

Typos
1. Pg8, L24, It should be: "to what extent can"
2. Pg 17, Table 17. A1, It should be "and start ....are recommended"
3. C1.1, I believe you mean a 50m deep mixed layer.

- Thanks, we have fixed these.